# TANGO2 binds crystallin alpha B and its loss causes desminopathy

Maike Stentenbach[1,2], Laetitia A. Hughes[1,2], Samuel V. Fagan [1,2], Blake Payne[1,2], Danielle L. Rudler[1,2], Stefan J. Siira [1,2], Tim McCubbin[3,4], Anaëlle Chopin[1,2], Kara L. Perks[1], Judith A. Ermer [2], James Hendry [1,2], Teagan S. Er[5], Shanti Balasubramaniam[6,7], Joel A. Eliades[8], Livia C. Hool [5,9], Nicolle H. Packer[10,11], Edward S. X. Moh[10,11], Benjamin S. Padman[1,2], Oliver Rackham [1,2,4,12,13] ✉ & Aleksandra Filipovska [1,2,8,13] ✉

Mutations in the *TANGO2* gene cause an autosomal recessive disorder characterised by developmental delay, stress-induced episodic rhabdomyolysis, and cardiac arrhythmias along with severe metabolic crises. Although *TANGO2* mutations result in a well characterised disease pathology, the function of TANGO2 is still unknown. To investigate the function of TANGO2, we knocked out the *TANGO2* gene in human cells and mice. We identify that loss of TANGO2 impairs intermediate filament structure, resulting in fragmented mitochondrial networks and formation of cup-like mitochondria. In male mice, loss of TANGO2 caused heart defects, reduced muscle function and glucose intolerance by remodelling of intermediate filaments, which altered the mitochondrial and cytoplasmic proteomes, N-glycosylation and nucleocytoplasmic O-GlcNAcylation. We identify that TANGO2 binds the small heat shock protein crystallin alpha B (CRYAB) to prevent the aggregation of the intermediate filament desmin and in the absence of TANGO2, mice develop desminopathy, which is consistent with features found in patients carrying mutations in either desmin or CRYAB.

Mutations in the transport and Golgi organization 2 (*TANGO2*) gene have been identified to cause a rare recessive genetic disorder known as TANGO2-deficiency disorder (TDD)[1–4]. TDD is a multisystemic disorder that can affect different organs including the brain, heart, and muscles, presenting as recurrent metabolic crises, muscle weakness, and neurological symptoms including developmental delay and seizures[5,6]. The role of TANGO2 in cells is still elusive, as well as the mechanisms by which *TANGO2* mutations cause TDD. Disease studies have suggested that TANGO2 may affect oxidative phosphorylation (OXPHOS), leading to reduced ATP production and increased oxidative stress[4,7]. Most recently, TANGO2 has been implicated in lipid metabolism and autophagy[8,9]. Furthermore, treatment with vitamin B5

[1]The Kids Research Institute Australia, Northern Entrance, Perth Children's Hospital, 15 Hospital Avenue, Nedlands, WA, Australia. [2]ARC Centre of Excellence in Synthetic Biology, University of Western Australia, Crawley, WA, Australia. [3]Australian Institute for Bioengineering and Nanotechnology, The University of Queensland, St Lucia, QLD, Australia. [4]ARC Centre of Excellence in Synthetic Biology, The University of Queensland, St Lucia, QLD, Australia. [5]School of Human Sciences, The University of Western Australia, Crawley, WA, Australia. [6]Genetic Metabolic Disorders Service, The Children's Hospital at Westmead, Sydney, NSW, Australia. [7]Discipline of Genomic Medicine, Sydney Medical School, University of Sydney, Sydney, NSW, Australia. [8]Department of Biochemistry and Molecular Biology, Monash Biomedicine Discovery Institute, Monash University, Melbourne, VIC, Australia. [9]Victor Chang Cardiac Research Institute, Darlinghurst, NSW, Australia. [10]ARC Centre of Excellence in Synthetic Biology, Macquarie University, Sydney, NSW, Australia. [11]School of Natural Sciences, Macquarie University, Sydney, NSW, Australia. [12]Curtin Medical School, Curtin University, Bentley, WA, Australia. [13]Curtin Medical Research Institute, Curtin University, Bentley, WA, Australia. ✉e-mail: oliver.rackham@curtin.edu.au; aleksandra.filipovska@uwa.edu.au

has shown some promise in vitro and in some patients[10–12], however, the mechanism of *TANGO2* mutations have not been shown to be directly linked or caused by a vitamin B5 deficiency in TDD related defects. Reports on the cellular localisation of TANGO2 in different organelles including Golgi, ER and mitochondria[2,3,5,8] have compounded the challenge in identifying the molecular function of TANGO2 and hence its contribution to disease.

Recently, the *C. elegans* and yeast homologues of TANGO2, known as haem-responsive gene 9 (HRG-9), were suggested to traffic haem between mitochondria and cell membranes[13]. However, the low binding affinity of TANGO2 for haem and the authors' suggestion that TANGO2 is not directly involved in mitochondrial export of haem but its loss results in mitochondrial accumulation of haem, indicates that the observed effects are a consequence of the genuine and yet unknown role of TANGO2[14]. Reduction in phosphatidic acid in *TANGO2* siRNA-treated cells[8], which is a precursor required for the biogenesis of mitochondrial phospholipid bilayers[15], may contribute to increased haem levels in mitochondria, making the function of TANGO2 elusive. Here, we knocked out TANGO2 in cells and in mice and using multiple omics technologies we evaluated the changes caused by loss of TANGO2. We identified that TANGO2 binds and activates CRYAB, which in turn prevents desmin aggregation. In the absence of TANGO2, reduced levels of desmin lead to desminopathy in mice and in patient cells with a *TANGO2* mutation.

## Results

### TANGO2 is important for maintenance of the mitochondrial network and OXPHOS biogenesis

We deleted *TANGO2* in CAL51 diploid human breast cells (*TANGO2^-/-^*) using CRISPR/Cas9 genome editing to identify its molecular role (Supplementary Fig. 1a, b). Loss of TANGO2 resulted in reduced levels of nuclear and mitochondria-encoded OXPHOS proteins (Fig. 1a), which has also been found in patients with *TANGO2* mutations[1,4,5]. The levels of mitochondrial DNA- and RNA-binding proteins that regulate OXPHOS expression were also reduced (Fig. 1b). As mitochondrial RNA-binding proteins were reduced[16–19], we measured de novo protein synthesis and found that the translation of the mtDNA encoded polypeptides (Fig. 1c) and de novo biogenesis of the OXPHOS complexes was reduced in the absence of TANGO2 (Fig. 1d). These defects were consistent with a reduction in oxygen consumption (Fig. 1e), that can be significantly rescued by plasmid-based TANGO2 expression (Supplementary Fig. 1c). We also observed reduced mitochondrial membrane potential (Fig. 1f), and a compensatory increase in mtDNA copy number (Fig. 1g) in the *TANGO2^-/-^* cells. Since OXPHOS and gene expression proteins were reduced we investigated the effects on the mitochondrial proteases. The mitochondrial matrix protease LonP1 was increased in *TANGO2^-/-^* cells, whereas the membrane proteases AFG3L2 and Yme1L1 were reduced, along with their targets OPA1 and CHCHD3 (Fig. 1h), possibly as a consequence of changes in the mitochondrial membranes in the absence of TANGO2[8]. BNIP3 and PINK1 levels were reduced, however the levels of LC3A/B, p62 and ATG5 were not affected by the absence of TANGO2 (Supplementary Fig. 1d), indicating that TANGO2 loss does not lead to autophagy. The mitochondrial morphology in the *TANGO2^-/-^* cells was fragmented (Fig. 1i), which was further exacerbated when the cells were grown in galactose (Fig. 1i), suggesting that TANGO2 is important for maintenance of the mitochondrial network, particularly during metabolic stress. Expression of wild-type TANGO2 rescued the mitochondrial network in the *TANGO2^-/-^* cells (Supplementary Fig. 1e). Although TANGO2 can co-localize with mitochondria, its cellular distribution is found throughout the cytoplasm, but not co-localized with Golgi, ER or peroxisomes (Supplementary Fig. 1f). Taken together, these data indicate that loss of *TANGO2* in human cells can compromise mitochondrial morphology that is important for the maintenance of the

reticular network that results in reduced translation and consequently decreased biogenesis and function of the OXPHOS system.

### Multi-omic profiling reveals the importance of TANGO2 for organelle and cytoskeletal structure and function

Metabolomic profiling identified significant reduction in most nucleotides, amino acids and some citric acid cycle metabolites (Supplementary Fig. 1g). The increase in NADH levels at the expense of decreased ATP, and reduced serine levels in the *TANGO2^-/-^* cells (Supplementary Fig. 1g), are consistent with impaired OXPHOS function that has been shown previously[20]. Noteworthy, are the increases in glucose 6-phosphate and fructose 6-phosphate that suggest glycolysis is required in response to reduced oxidative phosphorylation. TANGO2 loss caused a significant imbalance in cellular mRNAs, resulting in 3950 downregulated and 4152 upregulated genes, most notably resulting in the reduction of genes that regulate cytoskeletal and membrane structure and cell transport (Fig. 2a and Supplementary Data 1). However, the mitochondrial transcriptome was not significantly affected (Supplementary Fig. 2a, b), suggesting that defects in mitochondrial translation and biogenesis were a downstream consequence of TANGO2 loss and not a direct effect on mitochondrial gene expression. Gene ontology (GO) analyses showed that TANGO2 loss in cells grown in glucose affected membrane structures related to cell adhesion, enzyme function and glycosaminoglycan binding that are involved in extracellular matrix organization, response to wounding and cell structure (Fig. 2b). These changes were further exacerbated compared to control cells when the *TANGO2^-/-^* cells were grown in galactose (Supplementary Fig. 2c, d and Supplementary Data 1). Proteomic analyses showed that TANGO2 loss caused significant reduction in mitochondrial proteins related to OXPHOS, gene expression, biogenesis and proteostasis, Golgi and ER proteins involved in transport, folding and posttranslational modification such as N-glycosylation as well as cytoskeletal proteins (Fig. 2c, d and Supplementary Data 2). This suggests that TANGO2 may affect cytoplasmic processes that in turn can impact mitochondrial function, as mitochondria require interactions with the ER and cytoskeletal proteins for their dynamics, communication and segregation in cells.

Transcriptomic and proteomic analyses revealed defects in glycosylation pathways (Fig. 2a–d and Supplementary Data 1, 2), therefore we performed N-glycomic profiling, which revealed increased relative abundance of bisecting, α−2,6 sialic acid, core fucose and antenna fucose N-glycan features, at the expense of reduced oligomannose and α−2,3 sialic acid in *TANGO2^-/-^* cells compared to control cells (Fig. 2e and Supplementary Data 3). These glycan changes were entirely consistent with the identified glycogene changes identified in the transcriptomic analysis of *TANGO2^-/-^* cells (Fig. 2f and Supplementary Data 1); increased levels of MGAT3 responsible for bisecting; increased levels of FUT8 for core fucosylation; increased ST6GAL1, required for the synthesis of α−2,6 sialic acid; and FUT1, FUT3 and FUT4, required for antenna fucose synthesis. These analyses reveal that loss of TANGO2, while not a known glycogene, impacts the N-glycosylation status of proteins, which is known to modulate a variety of cellular processes, including transport and cell signalling[21].

### TANGO2 loss causes cardiomyopathy, desminopathy and metabolic dysfunction

To investigate the physiological relevance of TANGO2, we deleted it in mice (Supplementary Fig. 3a) and maintained the mice on a normal chow diet (NCD) or high fat diet (HFD) for 16 weeks. Echocardiography of the 20-week-old knockout mice on the NCD showed a significant increase in the fractional shortening and thickening of the intraventricular septum (IVS) accompanied by significant increase in the left ventricular posterior wall (LVPW) and decreased left ventricular internal diameter during diastole and systole (LVIDd and LVIDs, respectively) compared to control mice (Fig. 3a), consistent with the

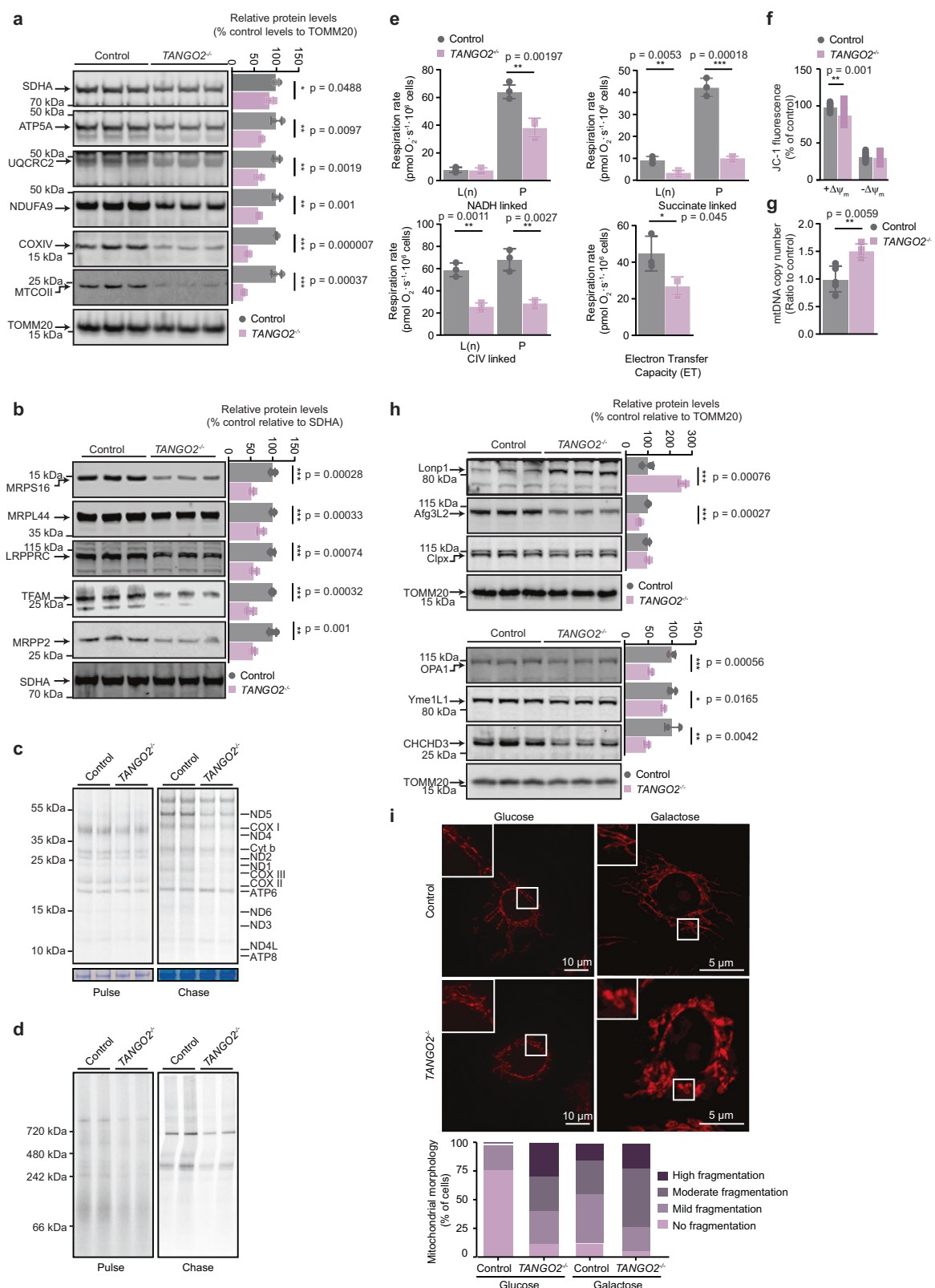

development of hypertrophy[22] that is also found in patients with *TANGO2* mutations[1,23]. The hypertrophy was further exacerbated in the *Tango2* [−/−] mice fed a HFD (Fig. 3a), showing cellular disarray (Supplementary Fig. 3b) and enlarged mitochondria with distorted cristae (Fig. 3b), typical of cardiomyopathy[24]. Loss of cytoskeletal architecture in the heart resembled a desminopathy, indicating the importance of TANGO2 for muscle tone. There was reduced size of specific fibres in

the skeletal muscle of 20-week-old knockout mice (Supplementary Fig. 3c) and changes in skeletal muscle mitochondrial morphology (Supplementary Fig. 3d), likely contributing to significantly reduced falling and reaching scores (Fig. 3c) and reduced ability to run on a treadmill (Fig. 3d), whereas general, exploratory behaviour was not affected (Supplementary Fig. 3e). These findings support a role for TANGO2 in cytoskeletal and organelle structure and function.

**Fig. 1 | *TANGO2* knockout decreases mitochondrial respiration and affects mitochondrial structure and morphology in vitro. a** Immunoblots of control and *TANGO2* $^{-/-}$ cells probed with total OXPHOS or COXII antibodies and normalised to TOMM20 ($n = 6$, biological replicates). **b** Immunoblots of proteins involved in mitochondrial translation (MRPS16, MRPL44, LRPPRC, TFAM, MRPP2) ($n = 6$, biological replicates). SDHA was used as a loading control. **c** De novo mitochondrial protein synthesis was measured by $^{35}$S methionine/cystine incorporation in control and *TANGO2* $^{-/-}$ cells. Coomassie stained gels are shown as loading controls ($n = 3$). **d** Measurement of de novo biogenesis of OXPHOS complexes by incorporation of $^{35}$S-labelled cysteine and methionine. The results in **c** and **d** are representative of three independent experiments. **e** Oxygen consumption through the N-linked, S-linked or CIV-linked pathway using either pyruvate/glutamate/malate, succinate or TMPD/ascorbate as substrates was measured for leak (L) and OXPHOS capacity (P) in control and *TANGO2* $^{-/-}$ cells grown in galactose media. ET capacity (ET) was measured using carbonyl cyanide p-trifluoromethoxyphenylhydrazone (FCCP) ($n = 4$, biological replicates). **f** Membrane potential measurements in control and *TANGO2* $^{-/-}$ cells in the presence and absence of FCCP ($n = 4$, biological replicates). **g** Mitochondrial DNA copy number in control and *TANGO2* $^{-/-}$ cells grown in glucose was quantitated by qPCR of *MT-CYB* and *HBB* ($n = 4$, biological replicates). Values for (**e**, **f** and **g**) are means ± SD. *$p < 0.05$, **$p < 0.01$ ***$p < 0.001$, Student's two-tailed $t$ test. **h** Steady state levels of proteins involved in mitochondrial dynamics (LONP1, AFG3L2, CLPX, OPA1, YME1L1 and CHCHD3) ($n = 6$, biological replicates). TOMM20 was used as a loading control. Relative abundance of proteins shown in panels (**a**, **b** and **h**) was analysed relative to the loading control ($n = 6$). Values are means ± SD. *$p < 0.05$, **$p < 0.01$ ***$p < 0.001$, Student's two-tailed $t$ test. **i** Mitotracker staining of control and *TANGO2* $^{-/-}$ cells grown in glucose and galactose media. Cells were stained with Mitotracker Orange prior to fixation and scored for fragmentation ($n = 100$ per cell line and treatment). Source data are provided as a Source Data file.

The *Tango2* $^{-/-}$ mice fed a NCD had a small but significant increase in their weight, however, when they were fed a HFD their weight was reduced by 20 weeks of age (Supplementary Fig. 4a). The *Tango2* $^{-/-}$ mice were glucose intolerant, developed insulin resistance (Supplementary Fig. 4b) and had increased circulating levels of insulin on both diets (Supplementary Fig. 4c). Haematoxylin and eosin staining revealed an increased accumulation of lipid droplets that was confirmed by oil red O staining (Supplementary Fig. 4d, e). Periodic acid-Schiff (PAS) staining revealed that loss of TANGO2 increased glycogen storage in the livers of the *Tango2* $^{-/-}$ mice fed a NCD and their mitochondria were round and swollen (Supplementary Fig. 4f, g). In contrast, glycogen levels were reduced in the *Tango2* $^{-/-}$ mice fed a HFD (Supplementary Fig. 4f). Dual energy x-ray absorptiometry (DEXA) measurements validated that the NCD fed *Tango2* $^{-/-}$ mice deposited more fat that contributed to their increased weight compared to controls (Supplementary Fig. 4h). These analyses show that TANGO2 loss alters organismal metabolism.

Proteomic analyses of heart, brain, liver and skeletal muscle tissues revealed specific changes caused by TANGO2 loss in proteins involved in energy metabolism, cytoplasmic and mitochondrial protein synthesis and vesicle-mediated transport were most significantly altered in the hearts and brains from the *Tango2* $^{-/-}$ mice on both diets (Fig. 3e–h, Supplementary Fig. 5a–j, Supplementary Fig. 6a, b and Supplementary Data 4). While the liver was affected the most on NCD and showed a total 2282 protein changes (1587 proteins were decreased and 695 proteins were increased), a HFD lead to the most protein changes in the heart with a total of 2293 proteins (1353 proteins were decreased and 940 proteins were increased) (Supplementary Fig. 5a, b). Changes in Golgi and ER proteins involved in posttranslational modifications were consistent with those identified in our *TANGO2* $^{-/-}$ cells and their effects on N-glycosylation in these cells. We investigated the nucleocytoplasmic O-GlcNAcylation of proteins by immunoblotting and identified a significant reduction in O-GlcNAcylation in the hearts, brains and skeletal muscle from *Tango2* $^{-/-}$ mice (Supplementary Fig. 6c–e) and increased O-GlcNAcylation in the livers of these mice compared to controls (Supplementary Fig. 6f). These data indicate that loss of TANGO2 impairs cellular architecture that impacts organelle morphology and function, in particular mitochondrial dynamics, protein synthesis N-glycosylation and O-GlcNAcylation, resulting in physiological and structural defects of organs reliant on energy that are consistent with the pathologies associated with *TANGO2* disease mutations.

## TANGO2 interacts directly with crystallin alpha B

To identify the molecular role of TANGO2 we used yeast two-hybrid screening to identify interaction partners of TANGO2. We found 121 out of 127 independent cDNA fragments identified in the screens that corresponded to full-length or near full-length CRYAB, encoding the small heat shock protein crystallin alpha B (Fig. 4a and Supplementary

Data 5). Mutations in CRYAB cause desminopathies resulting in similar cytoskeletal architecture and mitochondrial defects[25,26] as those found in the *Tango2* $^{-/-}$ mice. Introduction of a phosphomimetic mutation at the predominant phosphorylation site of CRYAB (S59E) did not affect its interaction with TANGO2 (Fig. 4b). However, a mutation that causes familial desminopathy (R120G), producing high molecular weight protein aggregates and loss of the chaperone activity of the protein in vitro, prevented the interaction between CRYAB and TANGO2 (Fig. 4b). Mutation of a structurally important cleft of TANGO2 (positions 25-27 converted to alanine, 25-27 A) eliminated its interaction with CRYAB, however, these mutations also reduced TANGO2 protein stability (Fig. 4b and Supplementary Fig. 7a). AlphaFold 3.0 modelling suggests that the interface interaction between CRYAB and TANGO2 occurs through their beta sheets (Fig. 4c). The interaction between TANGO2 and CRYAB is inherent to these two proteins, as purification of recombinant TANGO2 from bacterial cells efficiently co-purified recombinant CRYAB (Fig. 4d). Furthermore, we find that purified TANGO2 binds CRYAB with a greater affinity (Kd = $7.2 \times 10^{-6}$ M) compared to hemin (Kd = $4.3 \times 10^{-5}$ M) (Fig. 4e), that was previously suggested to bind TANGO2[13].

CRYAB binds desmin, an intermediate filament protein, and prevents its aggregation[27], enabling desmin to form scaffolds that connect mitochondria and the sarcoplasmic reticulum with contractile structures in muscle specific tissues[28]. Mutations in desmin or CRYAB lead to common molecular defects known as desminopathies that involve impaired mitochondrial morphology leading to cardiomyopathies, as a result of desmin aggregation or reduction in desmin and CRYAB[29–31]. TANGO2 loss reduced the levels of the intermediate filament protein vimentin in cells and its homologue, desmin, in heart and skeletal muscle, respectively (Fig. 4f), suggesting that TANGO2 is required for their stability in these tissues. CRYAB levels were not reduced in *TANGO2* $^{-/-}$ cells or hearts from *Tango2* $^{-/-}$ mice, but CRYAB was reduced in skeletal muscle of *Tango2* $^{-/-}$ mice (Fig. 4f). Mutations in the desmin gene (*Des*) result in desminopathies that manifest in common phenotypic changes[19] as those in the hearts and skeletal muscle of the *Tango2* $^{-/-}$ mice. The chaperone activity of CRYAB can remodel intermediate filament assembly by preventing their aggregation[32]. To determine if TANGO2 via its direct association with CRYAB is required for the stability of desmin under stress we carried out in vitro aggregation assays of desmin in the presence and absence of both TANGO2 and CRYAB at 22 °C, 37 °C and 44 °C. As previously reported, CRYAB reduces the aggregation of desmin at high temperatures, and here we show that the addition of TANGO2 to CRYAB increases its retention in the soluble fraction, preventing the aggregation of desmin and its shift into the pellets at 37 °C and even 44 °C (Fig. 4g and Supplementary Fig. 7b,c). TANGO2 also improves the solubility of CRYAB at higher temperatures (Fig. 4g and Supplementary Fig. 7b,c). We used fluorescent protein complementation[33] to show that loss of TANGO2 in cells reduced the interactions

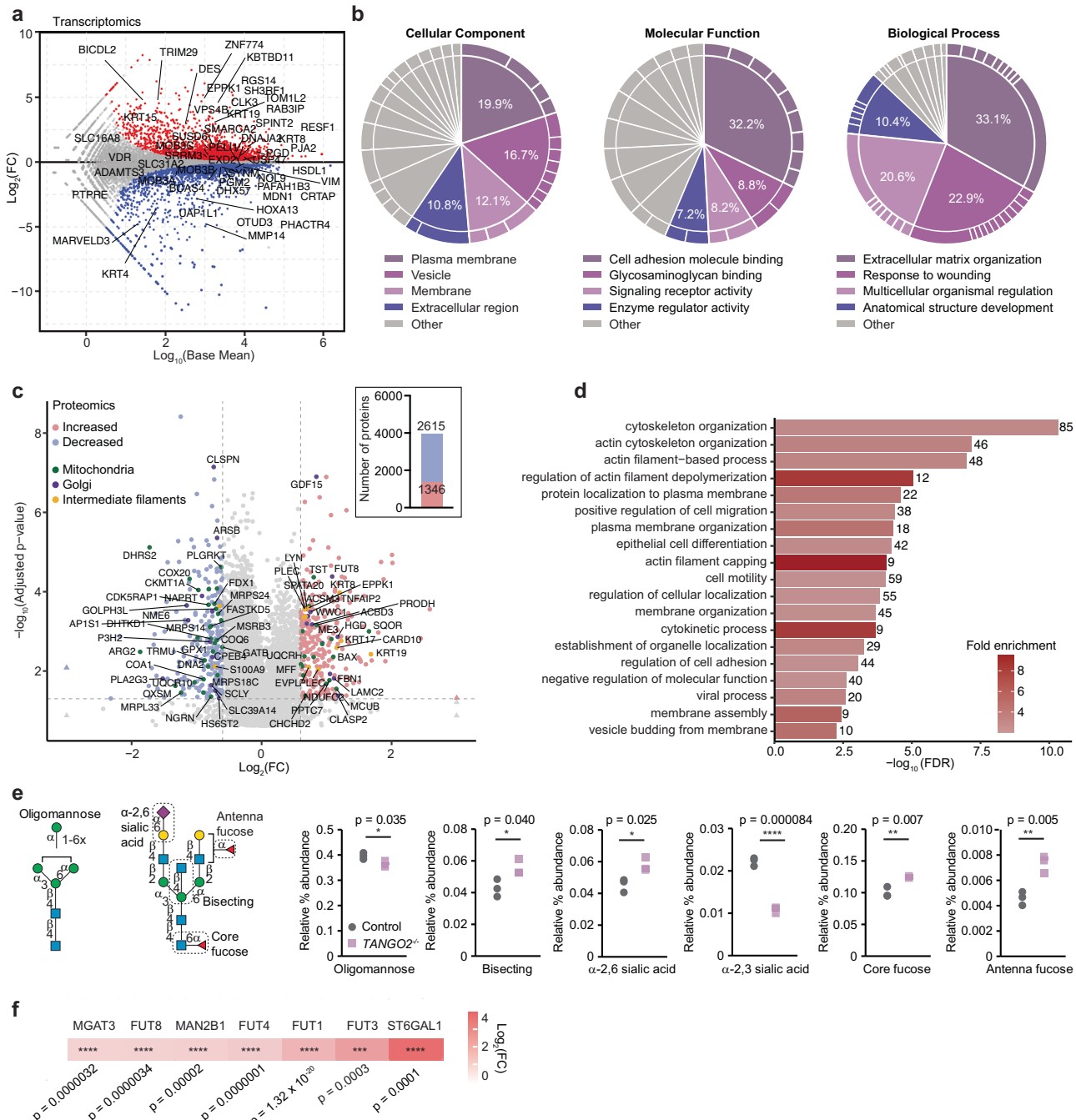

**Fig. 2 | Loss of TANGO2 affects cytoskeletal and membrane structure, protein transport and glycosylation. a** Transcriptome-wide changes in *TANGO2*⁻/⁻ cells compared with controls (*n* = 3). Increased genes are shown in red and decreased genes in blue. **b** GO analysis based on transcriptomic changes showing significantly changing pathways involved in biological processes, molecular function and cellular component in *TANGO2*⁻/⁻ cells compared with control cells. **c** Protein changes in *TANGO2*⁻/⁻ cells compared with control cells identified by mass spectrometry (*n* = 5 per genotype), the inset shows the number of significantly increased (red) or decreased (blue) proteins and adjusted *p* values are shown in Supplementary Data 2. Significantly increased and decreased proteins are shown in light red and light blue, respectively, mitochondrial proteins are shown in green, Golgi proteins are shown in purple and intermediate filament proteins are shown in yellow. **d** GO analyses based on proteomic changes show significantly changing reactome pathways in *TANGO2*⁻/⁻ cells compared with control cells. The results show the top 20 pathways with the highest −log₁₀(FDR) and the colour scale represents fold change (FC) for each pathway and set size is the number of genes within each pathway. **e** N-glycomic profile of control and *TANGO2*⁻/⁻ cells. **f** Changes in glycosylating enzymes identified by RNA-seq. The colour scale represents fold change (FC). All values shown in panels (**e** and **f**) are means ± SD *$p < 0.05$, **$p < 0.01$ ***$p < 0.001$, ****$p < 0.0001$, Student's two-tailed *t* test (*n* = 5 biological replicates).

between desmin and CRYAB, in a mode that is independent of the phosphorylation, glycosylation or common pathogenic mutations (Fig. 4h). We profiled CRYAB interactomes in the presence and absence of TANGO2 using proximity labelling and found that TANGO2 enables CRYAB association with cytoskeletal proteins, the majority of which associated with or were intermediate filaments (Fig. 4i). Importantly, we find that CRYAB self-association is reduced in cells lacking TANGO2, consistent with the in vitro data that show greater levels of CRYAB in the insoluble fraction in the absence of TANGO2 (Fig. 4g). Taken together, these data indicate that TANGO2

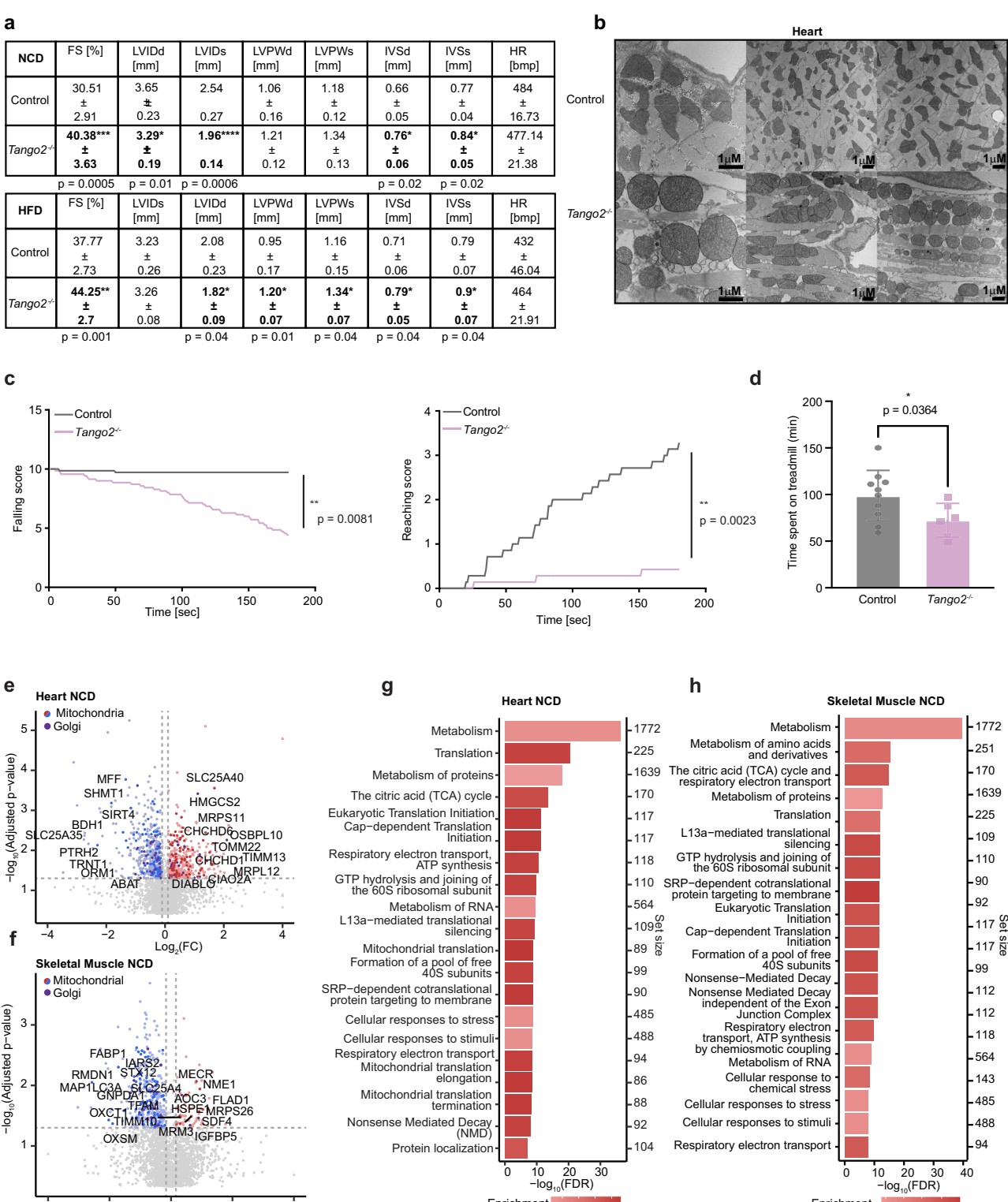

TANGO2 loss on mitochondria in the *TANGO2⁻/⁻* cells compared to control cells. Reconstructions of individual Mitotracker-

enhances CRYAB solubility and activity that is in turn required for intermediate filament assembly.

## TANGO2 modulates mitochondrial morphology and filopodia architecture

We used focused ion beam scanning electron microscopy (FIB-SEM) and three-dimensional reconstruction[33] to understand the structural effects of TANGO2 loss on mitochondria in the *TANGO2⁻/⁻* cells compared to control cells. Reconstructions of individual Mitotracker-

stained mitochondria from the *TANGO2⁻/⁻* cells identified smaller, fragmented and spherical mitochondria compared to control mitochondria (Fig. 5a and Supplementary Fig. 8). Morphometric quantitation of mitochondria from the *TANGO2⁻/⁻* cells identified reduced mitochondrial membrane curvature and an unusual increase in their negative curvature that manifests in a cup-like appearance (Fig. 5b). These data indicate that impaired intermediate filament structure fragments the mitochondrial network, causing changes in the curvature of mitochondrial membranes. In addition, we observed a dramatic

**Fig. 3 | *Tango2* deletion leads to cardiomyopathy and muscle weakness in vivo.**
**a** Echocardiographic parameters for 20-week-old control and *Tango2⁻/⁻* mice either fed NCD or HFD. LVIDd left ventricular internal diameter during diastole, LVIDs left ventricular internal diameter during systole, FS fractional shortening, LVPWd left ventricular posterior wall in diastole, LVPWs left ventricular posterior wall in systole, IVSd intraventricular septum in diastole, IVSs intraventricular septum in systole, HR heart rate ($n = 5$). All values are means ± SD, Student's two-tailed *t* test.
**b** Electron microscopy images of heart sections from 20-week-old control and *Tango2⁻/⁻* mice fed a NCD. The results are representative of at least three independent mice per genotype. **c** Hanging wire test in 20-week-old control and *Tango2⁻/⁻* mice. Falling and reaching score was determined after 3 min ($n = 7$).
**d** Involuntary treadmill exercise of 20-week-old control and *Tango2⁻/⁻* mice ($n = 7$

per genotype). All values are means ± SD *$p < 0.05$, **$p < 0.01$ ***$p < 0.001$, ****$p < 0.0001$ Student's two-tailed *t* test or Welch's *t* test for (**d**). Proteomic changes in (**e**) heart and (**f**) skeletal muscle from 20-week-old *Tango2⁻/⁻* mice fed a NCD compared to control mice ($n = 5$), and adjusted *p* values are shown in Supplementary Data 4. Significantly increased and decreased proteins were shown in red and blue, respectively, mitochondrial proteins are shown in green and Golgi proteins in purple. Gene ontology analyses show significantly changing reactome pathways in (**g**) heart and (**h**) skeletal muscle from 20-week-old *Tango2⁻/⁻* mice fed a NCD compared to control mice ($n = 5$). The results show the top 20 pathways with the highest $-\log_{10}(\text{FDR})$ and the colour scale represents fold change (FC) for each pathway and set size is the number of genes within each pathway.

disarray of the filopodia in *TANGO2⁻/⁻* cells (Supplementary Movie 1). Unlike the filopodia in wild-type cells, that radiate outward from the cell surface, those formed in the absence of TANGO2 crisscrossed extensively and in greater numbers (Fig. 5c and Supplementary Movie 1). This indicates that desmin/intermediate filament condensates in the absence of TANGO2 can nucleate and form extensive filopodia, consistent with in vitro observations where aggregates of desmin have been found to form desmin-rich protrusions[34]. Our findings are consistent with the pathological effects of the common pathogenic deletion of exons 3–9 in *TANGO2* patient cells that causes loss of TANGO2 and desmin aggregation (Fig. 5d) and consequent reduction of soluble desmin and increased levels of CRYAB (Fig. 5e). Our data show that TANGO2 is required for CRYAB activity and intermediate filament structure as its loss leads to a desminopathy that is consistent with those identified in patients carrying desmin and CRYAB mutations.

## Discussion

Here we identify that TANGO2 is a binding partner of the small heat shock protein CRYAB that enhances its activity, and identify its impact in tissues, cells and on organelle structure and function. Loss of TANGO2 caused a reduction of nuclear and mitochondria-encoded OXPHOS accompanied by a compensatory increase in mtDNA copy number, in response to the impaired OXPHOS function. Furthermore, loss of TANGO2 decreased mtDNA- and mtRNA-binding proteins that regulate mitochondrial gene expression, indicating a disruption in the coordination of mitochondrial gene expression. Reduction in OXPHOS complexes, oxygen consumption, and mitochondrial membrane potential align with previous reports of reduced OXPHOS function in patients with TANGO2 mutations[2,5]. The association of CRYAB with previously reported mitochondrial client proteins VDAC1 and CHCHD3[28] was altered in the absence of TANGO2. These changes indicate that TANGO2 is required for CRYAB function and consequently mitochondrial morphology, where loss of TANGO2 resulted in fragmented mitochondria that have reduced curvature and consequently form cup-like structures. Changes in mitochondrial curvature can be a consequence of impaired ether lipid metabolism affecting mitochondrial biogenesis caused by loss of different organelle genes, including TANGO2[33], whose loss we have shown in this study impaired mitochondrial biogenesis and cytoskeletal organisation. TANGO2 has been implicated in phospholipid metabolism and transport[8,35], suggesting further links between intermediate filaments, lipid exchange with mitochondria and lipid droplets[36–38], which in turn may explain the differential and non-exclusive cellular localisation of TANGO2[2,3,5,8,35].

Interestingly, TANGO2 loss caused significant imbalances in cellular gene expression, with downregulation of genes involved in cytoskeletal and membrane structure, as well as cell transport and defects in glycosylation pathways that resulted in changes of glycomic profiles. Changes in N-glycosylation including increased bisecting, sialylation and fucosylation have been implicated with changes to protein-protein interaction[39] and these may reflect the impaired

glycosylation of intermediate filaments that are aggregating in the absence of TANGO2. These findings indicate that TANGO2 plays a role in modulating the cytoskeletal environment, which can affect organelle function and dynamics, including mitochondria and Golgi, and these indirect effects can explain the consequences on altered lipid metabolism and haem transport identified previously[8,13]. At a physiological level, loss of TANGO2 caused metabolic perturbances that included glucose intolerance, insulin resistance, and increased circulating levels of insulin, particularly on a high-fat diet, that have also been identified in models of desminopathy[40] and muscular dystrophies[41] as well as type 2 diabetes found in patients suffering from desminopathy[42]. The most profound defects manifested as cardiomyopathy, liver steatosis, and impaired cytoskeletal architecture in the heart and skeletal muscle, that is also commonly found in patients with *TANGO2* mutations, including rhabdomyolysis[2,4–7]. Transcriptomic and proteomic analyses revealed that TANGO2 loss caused significant changes in energy metabolism, protein synthesis, cytoskeletal organisation and glycosylation enzymes that resulted in reduced O-GlcNAcylation and altered glycosylation profiles. These changes are consistent with defects in N-glycosylation and O-GlcNAcylation of intermediate filaments that have been reported to impact protein stability, mitochondrial function and cytoskeletal organisation cardiomyocytes and myocytes from patients[43–45].

The identification of CRYAB as an interaction partner of TANGO2 was key to identifying its molecular role and explains the phenotypic changes in the *TANGO2* knockout mice and patient cells that manifested as desminopathies, features that are found in diseases caused by mutations in *CRYAB* or *DES* and in mice lacking desmin[25,26]. We show that the pathogenic R120G mutation in CRYAB impairs its interaction with TANGO2, providing an additional molecular insight into the pathogenic mechanisms by which this mutation causes a desminopathy in patients. Therefore, TDD may be considered a subclass of desminopathy and TANGO2 mutations could be screened in the clinic when patients present with a desminopathy that is not caused by desmin or CRYAB mutations. The interaction between TANGO2 and CRYAB was found to be crucial for the stability of the intermediate filament proteins desmin and vimentin. We show that mutation of the TANGO2 cleft was important for its stability and thereby association with CRYAB, suggesting that this region is required for its function. These findings highlight the important role of TANGO2 in maintaining the cytoskeletal architecture that enables organelle networks, communication and function and have implications for future treatment options for patients with TANGO2 mutations, particularly identifying how vitamin B5 may help some of the symptoms of TDD patients.

## Methods

Our research complies with all relevant ethical regulations: Sydney Children's Hospitals Network Human Research Ethics Committee, Westmead, NSW, Australia (2019/ETH12990 and project number CCR2025/5) and by the Animal Ethics Committee of the UWA (2024/ET000129 and 2021/ET000274) and performed in accordance with

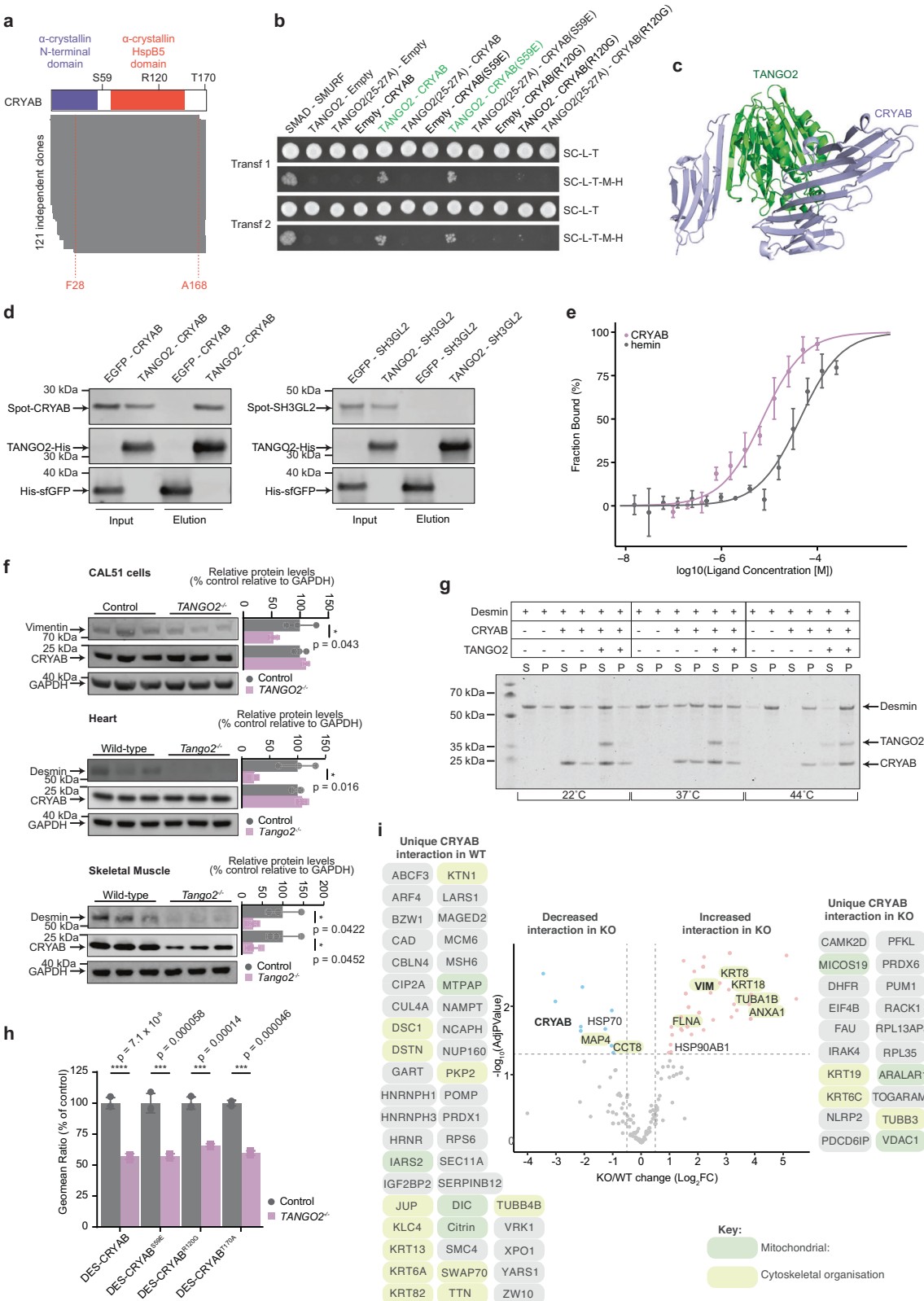

## Cell culture

Principles of Laboratory Care (NHMRC Australian code for the care and use of animals for scientific purposes, 8th Edition 2013).

### Cell culture

CAL51 (DSMZ, ACC-302) cells and TANGO2 (exons 3-9 *TANGO2* deletion) male patient fibroblasts (obtained from a skin arm tissue biopsy) were cultured and maintained at 37°C under humidified 95% air and 5%

$CO_2$ in Dulbecco's modified Eagle's medium containing 4.5 g/l glucose, 1 mM pyruvate, 2 mM glutamine, 10% foetal bovine serum (FBS) and 50 µg/ml uridine referred as high glucose media. Galactose media referred to Dulbecco's modified Eagle's medium deprived of glucose (DMEM, Gibco, Life Technologies) containing 10 mM galactose, and 10% FBS. Ethical approval for the patient fibroblasts was obtained from Sydney Children's Hospitals Network Human Research Ethics

**Fig. 4 | TANGO2 associates with CRYAB and regulates the protein-folding of desmin. a** Protein fragments of CRYAB that interact with TANGO2 identified from yeast two-hybrid screens of a human heart cDNA library. The common region found in different clones encoding CRYAB includes, at a minimum, amino acids 28-168. **b** Interaction of wild-type or mutant CRYAB and TANGO2 proteins in yeast two-hybrid assays. Interactions were assessed by survival on media lacking histidine and methionine (SC-L-T-M-H). The SMAD-SMURF interaction was used as a positive control. **c** AlphaFold model of the TANGO2 (green)−CRYAB (purple) interaction. **d** Confirmation of a direct interaction between recombinant proteins. SH3GL2 was used as a negative control. The results are representative of three independent experiments. **e** Protein binding affinity between TANGO2 and hemin (grey) or TANGO2 and CRYAB (pink) ($n = 3$ biological replicates). Values are means ± SD. **f** Immunoblots of CAL51 cells, heart and skeletal muscle lysates probed for CRYAB, desmin or vimentin. GAPDH was used as a loading control ($n = 3$ biological replicates). All values are means ± SD *$p < 0.05$, **$p < 0.01$ ***$p < 0.001$, Student's two-tailed $t$ test. **g** In vitro co-sedimentation assay of desmin, CRYAB and TANGO2. Filament assembly was initiated at 22 °C, 37 °C and 44 °C, and pellet (P) or aggregated fractions and supernatant (S) or soluble fractions analysed by SDS-PAGE and Coomassie Blue staining. Results are representative of three independent experiments. **h** Split-GFP protein-protein interaction assay for desmin and CRYAB variants (S59E, R120G and T170A) binding in control and TANGO2$^{-/-}$ cells ($n = 3$ biological replicates). All values are means ± SD ***$p < 0.001$, ****$p < 0.0001$, Student's two-tailed $t$ test. **(i)** BioID identification of CRYAB client proteins in control and TANGO2$^{-/-}$ cells ($n = 3$ biological replicates). Values are fold changes in abundance relative to adjusted $p$ values, listed in Supplementary Data 2; significantly increased or decreased binding proteins are shown in red and blue, respectively; unique client proteins in each of the cell lines are shown to the left and right of the volcano plot; mitochondrial and cytoskeletal organisation proteins are highlighted in green and yellow, respectively. Source data are provided as a Source Data file.

Committee, Westmead, NSW, Australia (2019/ETH12990 and project number CCR2025/5). Informed written consent was obtained for the tissue biopsy, use of cells for research and publication. All cells were STR profiled, tested and confirmed negative for mycoplasma contamination.

## Stable cell line generation
CAL51 cells were seeded at 60% confluency and transfected with 1.5 μg of a pD1311-AD mammalian Cas9 expression vector encoding a TANGO2-targeting gRNA using FuGENE HD (Promega) in Opti-MEM. Stable CAL51-TANGO2-EGFP cells were generated by transducing control cells with lentiviral particles produced by packaging pRRL-TANGO2-EGFP. At 72 h post transfection, single cells were sorted into 96-well plates based on GFP fluorescence signal using a FACSAria II (BD Biosciences) in PBS supplemented with 2% FBS. Disruption of *TANGO2* alleles was confirmed by Sanger sequencing performed by the Australian Genomic Research Facility (AGRF).

## Transient transfection
CAL51 cells were seeded at 60% confluency and transfected with 158 ng/cm2 of plasmid DNA for single-plasmid or 263 ng/cm2 for double-plasmid transfections using FuGENE HD in Opti-MEM. Experiments were performed 72 h post transfection.

## Mitochondrial isolation
Mitochondria were isolated from whole cell pellets after trypsinisation. The cell pellet was resuspended in 7 ml of mitochondrial swelling buffer (10 mM Tris Base, 10 mM NaCl and 1.5 mM MgCl$_2$, pH 7.5) and left to incubate on ice for a minimum of 30 min. Cells were homogenised using a glass homogeniser and 250 mM Sucrose T$^{10}$E$^{10}$ was added to the homogenised cells. Mitochondria were isolated by differential centrifugation, as described previously[46].

## Preparation of cell lysates
CAL51 cells were trypsinized and resuspended in a lysis solution containing 260 mM sucrose, 10 mM Tris-HCl, 100 mM KCl, 20 mM MgCl$_2$, 1% Digitonin, and complete EDTA-free protease inhibitor cocktail (Roche) for 30 min at 4 °C. Cells were centrifuged at max speed for 15 min at 4 °C to clarify lysates. Protein concentration was determined using a BCA assay.

## Immunoblotting
Specific proteins were detected using rabbit antibodies against LRP130 (sc66844, Santa Cruz Biotechnology; diluted 1:1000), MRPL44 (16394-1-AP, Proteintech, diluted 1:500), MRPS34 (HPA042112-100, Sigma, diluted 1:500), MRPS16 (16735-1-AP, Proteintech, diluted 1:1000), TFAM (HPA040648, Sigma, diluted 1:500), HSD17B10 (HPA001432, Sigma, diluted 1:500), HSP60 (ab137706, Abcam, diluted 1:1000), MTCO2 (ab198286, Abcam, diluted 1:1000), GAPDH (2118 s, Cell signalling, diluted 1:1000), NDUFAB (PA5-22191, Thermo Scientific, diluted 1:500), PINK1 (ab23707, Abcam, diluted 1:500), OXA1-L (21055-1-AP, Proteintech, diluted 1:1000), CPT II (0AAN00972, Aviva, diluted 1:1000), LONP1 (ab103909, Abcam, diluted 1:500), AFG3L2 (14631-1-AP, Proteintech, diluted 1:500), CLPX (HPA040262, Sigma, diluted 1:200), LC3A/B (12741, Cell Signalling, diluted 1:500), YME1L1 (ab170123, Abcam, diluted 1:1000), CHCHD3 (ARP57040, Aviva, diluted 1:500), ATG5 (ab108327, Abcam, diluted 1:1000), YY1 (ab109228, Abcam, diluted 1:500), CRYAB (15808-1-AP, Proteintech, diluted 1:500), Vimentin (3932, Cell signalling, diluted 1:1000), SH3GL2 (PA5-120939, Invitrogen, diluted 1:500) and TOMM20 (MA5-32148, Invitrogen, diluted 1:1000) and mouse antibodies against p62 (ab56416, Abcam, diluted 1:500), OPA1 (ab119685, Abcam, diluted 1:500) ATP5a (ab14748, Abcam, diluted 1:500), UQCRC2 (ab14745, Abcam, diluted 1:500), COX IV (ab14744, Abcam, diluted 1:500), BNIP3 (ab10433, Abcam, 1:1000) SDHA (ab14175, Abcam, diluted 1:1000), NDUFA9 (ab14713, Abcam, diluted 1:1000), GFAP (3670, Cell signalling, diluted 1:1000), Total OXPHOS Antibody Cocktail (ab110413, Abcam, diluted 1:1000), VDAC1/Porin (ab14734, Abcam, diluted 1:1000), Desmin (67793-1-Ig, Proteintech, diluted 1:1000) and O-GlcNAc (MA1-072, Invitrogen, 1:1000) in Odyssey Blocking Buffer (Li-Cor). GAL4 AD (135398, Abcam, diluted 1:4000), GAL4 DBD (sc 510, Santa cruz, diluted 1:500). IR Dye 800CW Goat Anti-Rabbit IgG or IRDye 680LT Goat Anti-Mouse IgG (Li-Cor) secondary antibodies were used, and the immunoblots were visualized using the Odyssey Infrared Imaging System (Li-Cor).

## Mitochondrial translation assays
*In organello* translation assays were carried out in control and knockout cells, as described previously[24,46,47]. Protein concentration was measured, and 30 μg of protein was resolved by SDS-PAGE and visualized by autoradiography.

## Blue native PAGE (BN-PAGE)
BN-PAGE was performed using isolated mitochondria from control and knockout cells, as described previously using 1% digitonin or 1% dodecyl-maltoside (DDM)[24].

## Mitochondrial respiration
Mitochondrial respiration was evaluated as specific O$_2$ consumption using the Oroboros O2k Oxygraph. 2 million cells either grown in high glucose or galactose media were permeabilised using 10 mg/ml Digitonin. Once oxygen flux was stabilised the O2k chambers were supplemented with substrates 10 mM glutamate/10 mM malate/ 5 mM pyruvate (Sigma), 10 mM succinate/ 0.5 μM rotenone (Sigma) or 0.5 mM TMPD/ 2 mM ascorbate (Sigma) to measure the non-phosphorylating resting state (LEAK state L(n)) in the NADH linked-, Succinate linked- or CIV-linked pathways. Addition of saturating adenosine diphosphate (ADP, Sigma) to the recording chamber, measured

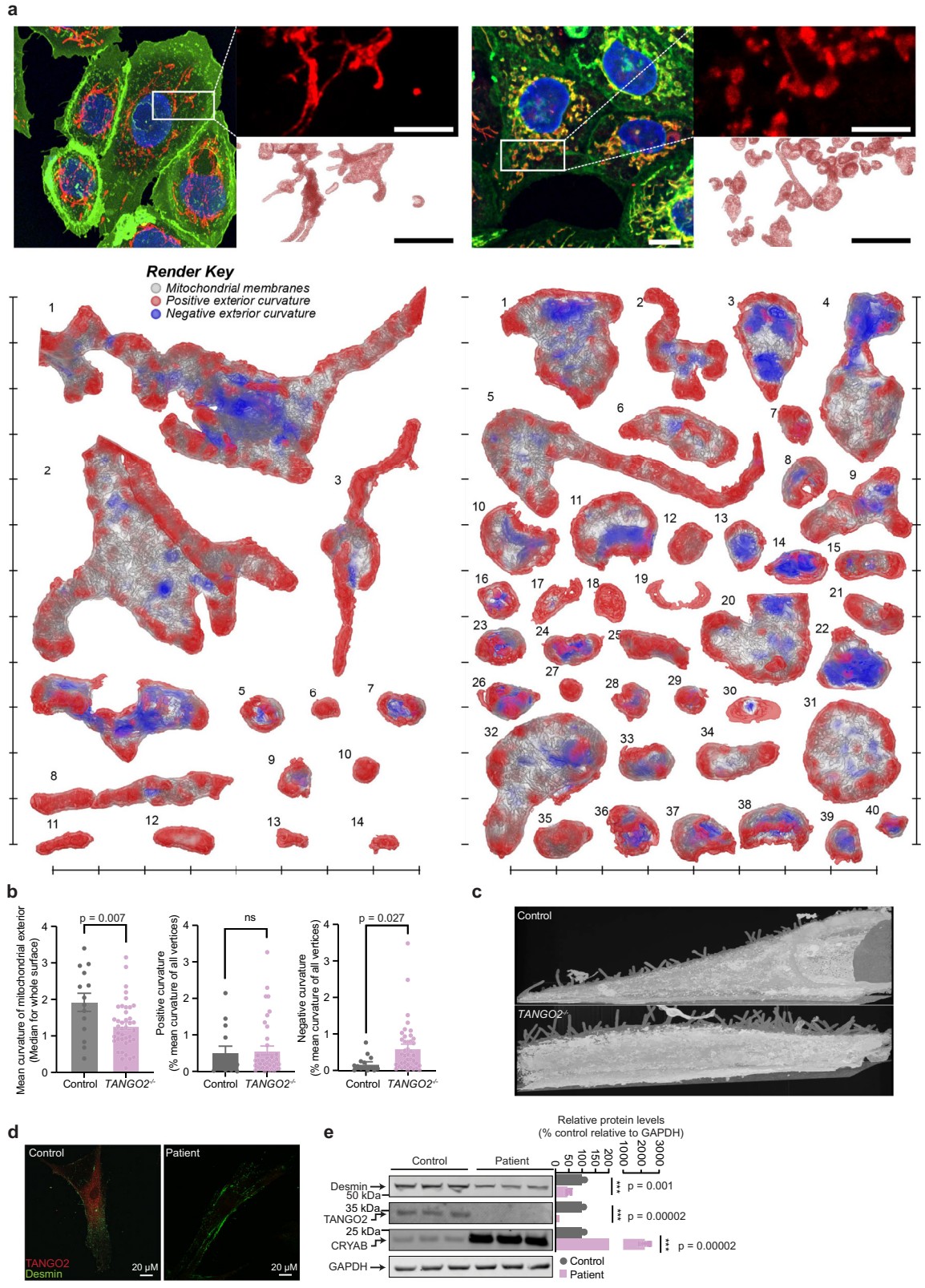

**Render Key**
- Mitochondrial membranes
- Positive exterior curvature
- Negative exterior curvature

OXPHOS Capacity (P) activity. Respiration was uncoupled by successive addition of carbonyl cyanide p-(trifluoro-methoxy) phenylhydrazone (FCCP) up to 3 μM to reach maximal respiration and measure the noncoupled electron transfer (E) capacity.

For evaluation of mitochondrial respiration in mice, mitochondria from liver and heart of 20-week-old wild-type and *Tango2⁻/⁻* mice were isolated and 150 μg were used. O2k chambers were

supplemented with substrates 10 mM succinate/ 0.5 μM rotenone (Sigma) to measure the non-phosphorylating resting state (L(n)) in the Succinate linked pathway. Addition of saturating ADP (Sigma) to the recording chamber, measured OXPHOS Capacity (P) activity. Respiration was uncoupled by successive addition of FCCP up to 2 μM to reach maximal respiration and measure the noncoupled electron transfer (E) capacity.

**Fig. 5 | TANGO2 is required for intermediate filament structure that is used by the mitochondrial network. a** Three dimensional renders of mitochondria detected in the control and *TANGO2*⁻/⁻ cells. For each cell, a full fluorescence micrograph, inset of the region isolated for FIB-SEM imaging, and the maximum value projection of raw AIVE data are shown; the reconstructed mitochondria and their distribution from the FIB-SEM dataset are shown. Scale bars, 2 μm (fluorescence micrographs) and 1 μm (fluorescence micrograph insets, electron micrographs and raw AIVE data); each notch around the reconstructed datasets represents 1 μm. The positive and negative curvature of mitochondria are shown in red and blue, respectively. **b** Quantitation of the mean curvature of the mitochondrial exterior. Values are means ± SD, *$p < 0.05$, **$p < 0.01$, Student's two-tailed *t* test ($n = 40$). **c** Sum projected images of 3D voxel values showing surface filopodia detected in the control and TANGO2⁻/⁻ cells. **d** Immunostaining of control and TANGO2 patient fibroblasts with desmin (green) and TANGO2 (red) antibody. The results are representative of three independent biological samples. **e** Immunoblots of control and TANGO2 patient fibroblasts probed with desmin, vimentin, TANGO2 and CRYAB. GAPDH was used as loading control ($n = 3$ biological replicates). All values are means ± SD *$p < 0.05$, **$p < 0.01$, ***$p < 0.001$, Student's two-tailed *t* test. Source data are provided as a Source Data file.

## Membrane potential measurement

12,500 cells were seeded in black 96-well plates and allowed to attach overnight. Cells were stained with 33 μM 5,5′,6,6′-tetrachloro-1,1′,3,3′-tetraethylbenzimidazolylcarbocyanine iodide (JC-1) (Molecular Probes, Thermofisher) in FBS-free DMEM and incubated for 60 min at 37 °C. Cells were incubated in 5% fatty acid free BSA (w/v) (Sigma) in PBS for five minutes at 37 °C, which was replaced with PBS. Control treatments were carried out with 50 μM FCCP added 10 min prior to JC-1 staining. Fluorescence readings were taken using a CLARIOstar (BMG Labtech). Data is presented as a ratio of 590 nm:520 nm values.

## DNA extraction and mtDNA copy number quantitative PCR

CAL51 cell DNA was extracted using a GeneJET Genomic DNA purification kit according to the manufacturer's instructions (Fermentas). Quantitative PCR was conducted on 100 ng of DNA using primers for cytochrome b, to determine levels of mitochondrial DNA, and primers for beta globin, to determine levels of nuclear DNA (IDT). Amplification was conducted using a Rotor-Gene Q (Qiagen) using SensiMix SYBR mix (Bioline).

## Fluorescence microscopy

Cells were plated onto 13 mm diameter glass cover-slips and allowed to attach overnight. Cells treated with 50 nM Mitotracker Orange (Molecular Probes, Thermo Fisher Scientific) for 15 min or Hoechst (Thermo Fisher Scientific) for 5 min. Alternatively, cells were transfected with a total amount of 1.5 μg of experimental plasmids and incubated for 72 h in normal growth conditions. The DeltaVision fluorescent microscope (GE Healthcare) was used for the acquisition of the images utilising a 60x or 100x objective oil immersion objective. Images were acquired using a CoolSNAP HQ CCD Camera and deconvolution was performed by softWoRx software (GE Healthcare). Mitochondrial morphology was quantified by scoring 50 different cells from two independent slides. Cells were classed into the following four categories: elongated (majority of the mitochondrial network was interconnected and had a reticular appearance); mildly fragmented (minor section of the network is fragmented but the majority of the network had a reticular, interconnected appearance); moderately fragmented (most of the network had a fragmented appearance but minor sections retain an interconnected structure); and highly fragmented (majority of the network was highly fragmented).

## RNA extraction and qRT-PCR

RNA was isolated from 2 million cells using the miRNeasy Mini kit (Qiagen) incorporating an on-column RNase-free DNase digestion to remove all DNA. cDNA was prepared using a Quantitect Reverse Transcription Kit (Qiagen). qRT-PCR were performed as described previously[48,49] for canonical mitochondrial mRNAs rRNAs and RNA19 normalized to 18S rRNA.

## RNA sequencing and differential expression analysis

RNA was isolated from CAL51 cells using a miRNeasy Mini Kit (Qiagen) incorporating an on-column DNase digestion according to the manufacturer's instructions. Libraries were constructed using an Illumina Stranded mRNA preparation kit and sequenced using the NovaSeq platform by AGRF. Sequenced reads were trimmed of adapter sequences with Trim Galore[50] 0.6.6 using Cutadapt 1.18[51] and FastQC 0.11.9[50] (--fastqc --nextera --clip_R1 1). Trimmed reads were quantified against the GENCODE v37 transcript sequences with custom mitochondrial transcripts (properly merged bicistronic transcripts and corrected terminal sequences) using Salmon 1.5.2[52] (-l A --seqBias --posBias --validateMappings). Transcript levels were summarised to gene-level with tximport[53] 1.14.2 and analysed for differential changes between wild type and knockout genotypes using DESeq2[54] 1.26.0 with independent hypothesis weighting (IHW) 1.14.0[55]. Gene expression changes with an adjusted *p*-value < 0.05 were considered significant. Plots were generated with ggplot2 3.3.5[56] and ggrepel 0.9.1[57] in R 3.6.3[58].

## Preparation of protein digests for proteomics analysis

Cells or tissues were lysed in lysis buffer (5% SDS, 50 mM triethylammonium bicarbonate), with tissue samples also disrupted via bead beating using an OMNI Bead Ruptor Elite pre-chilled to −1 °C and 5 rounds of 45 s pulses at 4 m/s with a 30 s dwell time. 200 μg of protein was dissolved in 50 μl cell lysis buffer and digested using S-trap Micro Spin columns (ProtiFi) as per the manufacturer's instructions. Briefly, dithiothreitol was added to a final concentration of 20 mM and incubated at 70 °C for 60 min. Proteins were alkylated by adding iodoacetamide to a final concentration of 40 mM and incubating at room temperature in the dark for 30 min. Proteins were acidified with 2.5 μl of 12% phosphoric acid and diluted with 165 μl of binding buffer (90% methanol, 100 mM final Tris). Samples were added to the S-Trap Micro Spin columns by centrifugation at 4000 *g* for 30 s then subsequently washed three times by successively loading 150 μl of binding buffer and centrifuging at 4000 × *g* for 30 s. Digestion was achieved by adding 1 μg sequencing-grade trypsin (Promega) and 25 μl of 50 mM ammonium bicarbonate and incubating overnight 37 °C. Peptides were eluted by successively adding 40 μl of 5% acetonitrile in 0.1% formic acid, 40 μl of 50% acetonitrile in 0.1% aqueous formic acid and 40 μl of 75% acetonitrile in 0.1% formic acid with a 30 s centrifugation step at 4000 × *g* between the addition of each elution buffer. The eluants were pooled, dried in a vacuum centrifuge and resuspended in 20 μl of buffer A (5% acetonitrile in 0.1% formic acid).

## Proteomics liquid chromatography and mass spectrometry

Samples were analysed using a Thermo Fisher Scientific Ultimate 3000 RSLC UHPLC and an Exploris mass spectrometer (Thermo Fisher Scientific) with FAIMS. Samples were injected on a reverse-phase PepMap 100 C18 trap column (5 μm, 100 Å, 150 μm i.d. x 5 mm) at a flowrate of 20 μl/min. After 2.7 min, the trap column was switched in-line with a Waters nanoEase M/Z Peptide CSH C18 resolving column (1.7 μm, 130 Å, 150 μm i.d. x 100 mm) and the peptides were eluted at a flowrate of 0.5 μl/min buffer A (5% acetonitrile in 0.1% formic acid) and buffer B (80% acetonitrile in 0.1 % formic acid) as the mobile phases. The gradient consisted of: 8% B for 0–4 min, 8–24% B for 4–47 min, 24–40% B from 47 to 50 min, 40–95% B from 50–54 min, 95% B for 54–57 min, followed by a wash, a return of 8% buffer B and equilibration prior to the next injection. The mass spectra were obtained in DIA mode with an MS1 resolution of 120,000, automatic gain control target set to

standard, maximum injection time set to auto and scan range from 400–1100 m/z. DIA spectra were recorded at resolution 30,000 and an automatic gain control target set to standard. Isolation windows schemas with a 1 m/z overlap and FAIMS settings were used as following: for the cell cultures; 70 isolation windows at 10 m/z each were used from mass 400–1100 with FAIMS settings at −45 V and −65 V respectively, and for mouse tissue; 10 m/z windows between mass 350–470, 5 m/z windows between 465–645 and 47 m/z windows between 640 and 1100 with fixed FAIMS at −52 V.

## DIA mass spectrometry data analysis

Data analysis was performed with Spectronaut version 16 (16.3.221108.53000) using direct DIA analysis and default settings unless otherwise specified[59]. Briefly, spectra were searched against either the *Homo sapiens* proteome database, downloaded from UniProt 09/09/2020, or the *Mus musculus* database, downloaded 25/02/2021, for the respective cell culture and mouse model studies. Carbamidomethylation was set as a fixed modification and methionine oxidation and N-terminal acetylation as variable with 1% false discovery rate (FDR) cut-offs at the peptide spectral match, peptide and protein group levels. Quantitation was performed at the MS2 level with Q-value data filtering and cross run normalization with automatic row selection and without imputation. A protein group specific proteotypicity filter was used. Proteins were considered significant if they had a false discovery rate of less than or equal to 5% and an absolute log2-fold change equal to or greater than 0.6. Volcano plots were made using ggplot2[56] in R[58] and custom genes were highlighted based on ontology results from Spectronaut. GO analysis was performed using PANTHER 17.0[60] for reactome pathways.

## Glycomics sample preparation

Frozen cell pellets containing 2 million cells were chemically lysed (0.1 M Tris-HCl pH 7.6, 0.1 M DTT, 2% SDS) with heat (56 °C, 1 h, 200 rpm shaking). Following cell lysis, protein concentration was determined using the Qubit protein assay (Thermo Fisher) according to the manufacturer's instructions. An aliquot equivalent to approximately 400,000 cells were alkylated with 0.2 M iodoacetamide (1 h, RT, dark). Samples were concentrated to 10 μl in volume under vacuum and immobilised on activated PVDF membrane as described[61]. Membrane spots were excised, transferred into a 96-well plate and blocked with 1% (w/v) polyvinylpyrrolidone (40 kDa, linear) in methanol, and washed with water twice before *N*-linked glycomics analysis.

## N-glycan release and analysis

*N*-glycan sample preparation was performed as described[61] with some modifications. In brief, *N*-glycans were first released via incubation with 5 U of PNGase F (Promega) in each sample well at 37 ° C for 16 h. Released *N*-glycans were obtained, then deamidated with ammonium acetate pH 5 before drying under vacuum. Glycans were then reduced using 1 M sodium borohydride in 50 mM potassium hydroxide at 50 °C for 3 h. Reduced N-glycans then diluted 6-fold using deionised water and neutralised using glacial acetic acid before loading onto carbon (Supelclean ENVI-carb) packed tips for desalting accordingly[62]. Purified glycans were injected onto a Hypercarb Porous Graphitized Carbon (PGC) column (3 μm, 1 mm × 30 mm) using Agilent 1260 HPLC coupled to a ThermoFisher Velos Pro ion trap at a flow rate of 20 μl/min with the following gradient parameters: Buffer A: 10 mM ammonium bicarbonate, Buffer B: 70% acetonitrile with 10 mM ammonium bicarbonate,: 0–3 min–0% B, 4 min–14% B, 40 min–40% B, 48 min–56% B, 50–54 min–100% B, 56–60 min–0% B. Glycans were detected in negative mode, with a m/z acquisition window of 400–2000[62]. Glycan composition was identified manually using monoisotopic mass and MS/MS fragmentation, isomers of known glycans assigned according to PGC retention time rules[62] MS peak area quantitation was performed using Skyline[63].

## Cell culture intracellular metabolomics sample preparation

In total, 500 μL ice cold 50% methanol was added to pellets of 1 million snap-frozen cells and the mixture was mixed, thawed on ice and sonicated for 5 min in an ice bath. 200 μL of 0.5 mm silica glass beads were added to samples which were loaded onto an OMNI Bead Ruptor Elite that was pre-chilled to −1 °C. Metabolites were extracted using 5 rounds of 45 s pulses at 4 m/s with a 30 s dwell time. Metabolites were purified by adding 250 μL chloroform, which was then vortexed and centrifuged at 16,000 RCF for 5 min at 4 °C. The supernatant was recovered, and the organic phase reduced using a vacuum concentrator until approximately 200 μL remained. Samples were further diluted with 500 μL milliQ water, freeze-dried and reconstituted in 100 μL of 2% acetonitrile where they were then split: 40 μL for amino acid analysis and 60 μL for central carbon metabolite quantification.

## Central carbon metabolite quantification

Samples were spiked with 2.5 μM azidothymidine as an internal standard and metabolites were analysed using liquid chromatography–tandem mass spectrometry (LC-MS/MS) performed using a Shimadzu Nexera X2 UHPLC system coupled to a Shimadzu 8060 triple quadrupole mass spectrometer. Chromatographic separation was achieved on a Phenomenex Gemini NX-C18 column (00A-4453-B0) with guard column (SecurityGuard Gemini NX-C18, 4 × 2 mm), operated at 45 °C at a flow rate of 300 μL/min. Mobile phase A was 7.5 mM tributylamine aqueous solution (pH to 4.95 with acetic acid), and mobile phase B was acetonitrile. Mass spectrometry was achieved using the scheduled multiple reaction monitoring (sMRM) method on the negative ionization mode using transient ions as previously published[64]. Concentrations of each metabolite were calculated based on standard curves from serial dilutions of authentic chemical standards (Sigma). Collected data were processed using LabSolutions InsightTM (Shimadzu).

## Intracellular amino acid analysis by HPLC

Samples were mixed 1:1 with an internal standard of 500 μM sarcosine and 250 μM 2-aminobutanoic acid. Amino acids were derivatized as previously described[65] and analysed on a Vanquish Core UHPLC (Thermofisher Scientific, Waltham, Massachusetts, USA) equipped with a Vanquish Fluorescence detector and high performance autosampler with online derivatisation.

The derivatised amino acids were separated on a Zorbax Extend C-18 column (150 ×4.6, 3.5 μm ID; Agilent Technologies Inc. Santa Clara, California, USA) protected by a Gemini C18 SecurityGuard Column (Phenomenex, Torrance, California, USA) kept at 37 C. Mobile phase A was 40 mM $Na_2HPO_4$ containing 0.02% $NaN_3$ (w/v; pH 7.8) and mobile phase B was methanol/acetonitrile/water (45:45:10 v/v/v). The separation was achieved by the following gradient: 2% B isocratic for 1 min, 2%-30% B from 1-14 min, 30–25% B from 14 to 15 min, step increase to 40%, 40–45% B from 15.1 to 18 min, increase to 60% B from 18 to 20 min, followed by a cleaning and equilibration step to recondition the column to the original conditions.

Amino acids were monitored using a fluorescence detector with $340_{ex}$ and $450_{em}$ nm for OPA derivatised compounds and $262_{ex}$ and $305_{em}$ for FMOC derivatised compounds. Quantification was based on external calibration curves of an amino acid mix with individual concentrations between 500 μM to 2 μM for OPA derivatised compounds or 1000 μM to 4 μM for FMOC derivatised compounds, respectively.

## Metabolomics data analysis

Metabolomics data were compiled and analysed using MetaboAnalyst 5[66]. When peaks were detected but below the standard curve, the standard curve was extrapolated to estimate concentrations, otherwise missing values were imputed by 1/5 of the minimum detected concentration. Concentrations were $log_{10}$-transformed and differentially abundant metabolites were identified using one-way ANOVA,

with post-hoc tests using the Fisher's least significant difference method to identify differential comparisons between groups for that metabolite.

## Animals and housing

Heterozygous *Tango2* knockout mice on a C57BL/6 N background were generated by the Monash Genome Modification Platform (MGMP) and the Australian Phenomics Network (APN). The knockout of the *Tango2* gene was introduced by CRISPR technology. Male age- and littermate-matched (10-week-old and 20-week-old) wild-type and knockout mice (*Tango2*) were housed in standard cages (45 cm × 29 cm × 12 cm) under a 12-h light/dark schedule (lights on 7 a.m. to 7 p.m.) in controlled environmental conditions of $22 \pm 2\,°C$ and $50 + 10\%$ relative humidity. Mice were provided normal chow diet (NCD) (Rat and Mouse Chow, Specialty Feeds, Perth, Australia) and water ad libitum. High fat diet (Rat and Mouse Chow, Specialty Feeds, Perth, Australia) was provided ad libitum for 14 weeks. Dual energy x-ray absorptiometry was used to analyse the body composition of 20-week-old control and *Tango2*[-/-] mice. This study was approved by the Animal Ethics Committee of the UWA and performed in accordance with Principles of Laboratory Care (NHMRC Australian code for the care and use of animals for scientific purposes, 8th Edition 2013).

## Metabolic studies

Six-week-old male mice were fed either NCD or HFD for 14 weeks while their body weight was monitored weekly. Intraperitoneal glucose tolerance test (GTT) and insulin tolerance test (ITT) were performed at 18 weeks (for GTT) and 19 weeks (for ITT) as previously[67]. Mice were euthanized by cervical dislocation at 20 weeks of age after 14 weeks on their respective diets.

## Enzyme-linked immunosorbent assays

Serum from fasted mice was analysed for circulating insulin levels using an insulin enzyme-linked immunosorbent assay (ELISA) kit (EZRMI-13K; Millipore) and performed according to the manufacturer's protocol.

## Histology

Heart, liver and skeletal muscles were fixed with 10% neutral buffered formalin and processed as described previously[67]. Hematoxylin and eosin and oil red O staining was performed as described in ref. 68. Periodic Acid-Schiff (PAS) stain (Sigma-Aldrich) was performed on paraffin embedded livers according to the manufacturer's protocol.

## Echocardiography measurements

M-Mode echocardiographic studies were performed on mice under light methoxyflurane anaesthesia with the use of an i13L probe in the long axis on a Vivid 7 IQ ultrasound system (GE Healthcare, Little Chalfont, UK) as previously described[69]. Each *n* represents the average of three quantitative measurements from five wild-type and five *Tmango2*[-/-] mice for each treatment group.

## Perfusion and electron transmission microscopy

20-week-old male mice were perfused with Ringer's solution, followed by 30–50 ml 4% paraformaldehyde in 0.2 M phosphate buffer and 30 ml of 2.5% glutaraldehyde in 0.2 M phosphate buffer. Liver, heart and skeletal muscle were dissected and stored in phosphate-buffered 2.5% glutaraldehyde for seven days. Tissues were cut in 50–100 mm sections in the following region: apex region of the heart, left lobe of the liver, longitudinal cut of the skeletal muscle. Processing and imaging were performed as described previously[17].

## Open field test

All behavioural tests were carried out on 20-week-old male mice. Prior to each experiment mice were taken to the testing room in their home cages and given 10 min to habituate. All tests were carried out in the 'light' phase of the light-dark cycle. To test general activity and locomotor ability mice were placed into a $30 \times 30$ cm box, under a constant light source. The floor of the box was divided into a $4 \times 4$ square grid, which was used to track their movement throughout the box. The number of boxes crossed by the mice over a 10 min period was measured and the time mice spent in the inside boxes.

## Exercise endurance test

An exercise endurance test used the Exer 3/6 multi-lane treadmill. One - two days prior to the exercise stress test, mice were acclimatised to treadmill running by performing 10 min of exercise at a speed of 10 m/min (0–45% incline). On the day of the experiment, each mouse was placed on the treadmill and for a 10 min period with the shock stimulus being a group of metal bars/grill located at the back of the treadmill turned on (163 V; 1.5 mA). Mice are exposed to a shock if they decide to exit the treadmill running belt and sit on the metal bars/grill. Once the shock stimulus has been turned on the treadmill was turned on to allow mice to commence running at 10 m/min. Running speed was increased by 2.5 m/min every 10 min until a speed of 20/min has been reached at which point mice were left to run at this speed until exhaustion. Exhaustion is defined as the time point whereby mice remain at the back of the treadmill on the shock pad/stimulus for $\geq 5$ s or receive 4 continuous shocks whichever comes first. Upon reaching exhaustion mice are removed from the treadmill and returned to their home cage.

## Tissue homogenates

Tissue pieces (3 mm by 3 mm) from liver, heart and brain were homogenised in 200 µl of CEB buffer (100 mM tris, 2 mM $Na_3VO_4$, 100 mM NaCl, 1% Trito X-100, 1 mM EDTA, 10% glycerol, 1 mM EGTA, 0.1% SDS, 1 mM NaF, 0.5% deoxycholate, 20 mM $Na_4P_2O_7$, pH 7.4) containing phosSTOP Phosphatase Inhibitor Cocktail (Roche) and EDTA-free Complete Protease Inhibitor Cocktail (Roche) using a bead beater. The homogenates were centrifuged at 9000 x *g* for 5 min at 4 °C and repeated until a clear tissue homogenate was obtained. The protein concentration was quantified using the bicinchoninic acid (BCA) assay.

## Yeast two-hybrid screening

The full coding sequence of human TANGO2 (aa 1–276, GenBank accession number NM_001283106.3) was cloned as an N-terminal fusion to the LexA DNA-binding domain in plasmid pB29 or with the Gal4 DNA-binding domain in plasmid pB66 (Hybrigenics Services). Yeast two-hybrid screening was performed using cDNA libraries derived from human heart (adult ventricle and embryo heart) and 221 million and 81.9 million independent transformants were screened using LexA and Gal4 fusions, respectively (Hybrigenics Services). The coding sequences of positive clones were amplified by PCR and Sanger sequenced to identify interacting proteins. Interactions between wild-type and mutant CRYAB and TANGO2 proteins were examined in pairwise combinations, where CRYAB was codon optimized for yeast expression and subcloned into pP14 (Hybrigenics Services) for expression under the control of the *MET25* promoter. Interaction pairs were tested as two independent clones for each interaction and serial dilutions (undiluted, $10^{-1}$, $10^{-2}$, $10^{-3}$) of yeast cells (culture normalized to $5 \times 10^7$ cells) expressing both bait and prey fusions were spotted on media lacking tryptophan and leucine, as a growth control and to verify the presence of both the bait and prey plasmids. The serial dilutions were also spotted onto selective medium without tryptophan, leucine, histidine and methionine, to test for interaction-dependent activation of the *HIS3* reporter gene.

## Protein purification

CRYAB, TANGO2 and SH3GL2 open reading frames were codon optimised for expression in *Escherichia coli*, assembled from overlapping oligonucleotides and sub-cloned into pET-24a( + )-His-sfGFP. TANGO2 was expressed as a C-terminally His-tagged fusion protein, while CRYAB and SH3GL2 were expressed as N-terminally Spot-tagged proteins in *E. coli* BLR(DE3) cells (Novagen) by induction with IPTG at room temperature for 16 h. Cells were lyzed by sonication in 50 mM Tris-HCl (pH 8.0), 0.3 M NaCl, 5 mM imidazole and 1x cOmplete Protease Inhibitors (Roche). Lysates were then clarified by centrifugation and incubated for 1 h with His-Select nickel affinity beads (Sigma) at 4 °C. Protein bound beads were then transferred to BioRad Poly-Prep columns washed with 50 mM Tris-HCl (pH 8.0), 0.3 M NaCl and 5 mM imidazole. Proteins were then eluted with 50 mM Tris-HCl (pH 8.0), 0.3 M NaCl, 250 mM imidazole, for His -tagged proteins, or 50 mM Tris-HCl (pH 8.0), 0.3 M NaCl, 5 mM imidazole and 0.5 mM Spot peptide, for Spot-tagged proteins. Eluted proteins were then dialyzed into 25 mM HEPES (pH 8.0), 20 mM NaCl, pH 8.0, 1 mM DTT at 4 °C using Amicon Ultra 10 kDa spin columns. Protein concentration was determined by the bicichroninic acid (BCA) assay using bovine serum albumin (BSA) as a standard.

## Bacterial protein interaction assays

*E. coli* BLR(DE3) cells were co-transformed by heat shock with either pET24(+)-Spot-CRYAB and pET21a(+)-TANGO2-His, or pET24(+)-Spot-CRYAB and pET24(+)-His-sfGFP. Co-transformants were inoculated into LB media containing 50 μg/mL kanamycin and 100 μg/mL ampicillin and induced with 0.5 mM IPTG at 22 °C for 18 h. Cells were harvested by centrifugation and lysed by sonication in 50 mM Tris-HCl (pH 8.0), 0.3 M NaCl, 5 mM imidazole and 1x cOmplete Protease Inhibitor Cocktail (Roche). Lysates were clarified by centrifugation at 11,000 × g for 45 min at 4 °C, then added to His-Select nickel affinity beads (Sigma) and incubated for 45 min at 4 °C. Beads were washed with 50 mM Tris-HCl (pH 8.0), 0.3 M NaCl and 10 mM imidazole in a BioRad Poly-Prep column, protein was then eluted with 50 mM Tris-HCl (pH 8.0), 0.3 M NaCl and 250 mM imidazole. Aliquots of total soluble protein and protein eluates were analyzed by western blotting with anti-GFP (Novus, NB600-308), anti-c22orf25 (TANGO2) (Proteintech, 27846-1-AP), anti-Alpha B Crystallin (Proteintech, 15808-1-AP) and anti-SH3GL2 (Invitrogen, PA5-120939).

## MST binding measurements

Pure His-tag proteins were labelled using RED-tris-NTA dye with the Monolith His-Tag Labelling RED-tris-NTA 2nd Generation kit following manufacturer's instructions (NanoTemper Technologies). Briefly, 200 nM of protein was combined with RED-tris-NTA dye and incubated for 30 min at room temperature. Samples were centrifuged at 4 °C for 10 min at 15,000 g and the supernatant was retained. MST measurements were performed on a Monolith NT.115 (NanoTemper Technologies) using MO.Control v1.6.1 software (NanoTemper Technologies). 10 mM of hemin was dissolved in DMSO and further diluted to 1 mM in water to remove the impact of organic solvents on protein interactions, following MO.Control software instructions. For each assay, 100 nM of RED-tris-NTA labelled TANGO2 was loaded into each capillary with an equal volume of either 15 μM of pure CRYAB protein or 1 mM hemin. MO.Affinity Analysis v2.3 (NanoTemper Technologies) was used for MST binding analysis.

## Aggregation assay

Human recombinant desmin (Progen, 62016) was reconstituted to 1 mg/ml in 30 mM Tris-HCl pH 8, 9.5 M urea, 2 mM DTT, 2 mM EDTA, 10 mM methylammonium chloride, then diluted to 0.25 mg/ml with 10 M Urea, 5 mM Tris-HCl pH 8.4, 1 mM EDTA, 0.1 mM EGTA, 1 mM DTT. Urea concentration was reduced stepwise (6 M, 4 M, 2 M and 0 M) via dialysis against 5 mM Tris-HCl pH 8.4, 1 mM EDTA, 0.1 mM EGTA,

1 mM DTT with at least 4 h allowed for each dialysis step. Desmin filament aggregation was assessed as previously described[70]. Filaments were assembled at 0.1–0.15 mg/mL desmin with or without an equimolar amount of CRYAB, or with equimolar amounts of both CRYAB and TANGO2. Assembly was carried out at 22 °C, 37 °C, or 44 °C by adding 20X assembly buffer to achieve final solution concentrations of 22.5 mM Tris-HCl pH 8.4, 50 mM NaCl, 1 mM DTT. Samples were incubated at specified temperatures for 16 h prior to centrifugation at 2500 x g for 10 min. Supernatants were removed, and both fractions were analysed by SDS-PAGE and visualised with Coomassie blue stain.

## Flow cytometry measurements of protein association using split-GFP

Control and *TANGO2*$^{-/-}$ cells were co-transfected using 158 ng/cm$^2$ of plasmid DNA with pGFP1-10-DES and either pCRYAB-GFP11, pCRYAB-S59E-GFP11, pCRYAB-R120G-GFP11, or pCRYAB-T170A-GFP11, alongside pMem-mCherry at a 9:1 ratio as described previously[33], and incubated for 48 h. The cells were then trypsinized and the fluorescent signals of GFP and mCherry were measured in PBS + 2% FBS (vol/vol) using a Cytek Aurora system (Cytek Biosciences). Data are expressed as the ratio of GFP and mCherry intensity geometric means (GFP:mCherry) of GFP and mCherry double-positive cells. Changes in fluorescent complementation between proteins were analyzed as previously described[33].

## Proximity labelling of CRYAB interaction networks using biotinylation identification (BioID)

Wild-type or *TANGO2* knockout CAL51 cells were transduced with lentiviral particles produced by packaging plasmids designed to express a TurboID tag fused to the C-terminus of CRYAB or a cytoplasmic TurboID tag. Cells were grown as a monolayer in 15 cm dishes and labelled using 500 μM biotin for 10 min prior to harvesting. Lysates were prepared, biotinylated proteins enriched, on-bead trypsin digestion of biotinylated proteins and mass spectrometry performed as described in Branon et al.[71]. Mass spectrometry data was acquired using Thermo QExactive Orbitrap with an Ultimate 3000 RSLCnano HPLC. Peptides were resolved over a linear gradient 2% to 37% (v/v) acetonitrile with 0.1% formic acid over 35 mins, followed by column re-equilibration. Top 10 most abundant ions were isolated for MS/MS with a normalized collision energy of 30 eV. Protein identification and peak area quantitation was performed using Proteome Discoverer 2.5, with modifications to include cysteine carbamidomethylation (static), asparagine/glutamine deamination (variable) and lysine biotinylation (variable).

## SEM microscopy and 3D analyses

Control and *TANGO2*$^{-/-}$ cells were cultured for 48 h in 35 mm Ibidi 500-grid plastic-bottomed μ-dishes before staining with 50 nM MitoTracker Orange (5 min), and primary fixation with phosphate-buffered 4% paraformaldehyde at 37 °C for 1 h. The fixed samples were post stained (30 min) with 2.5 mg/ml CellMask Deep Red (Thermo Fisher Scientific) and 1 μg ml$^{-1}$ Hoechst 33342 (Merck) for room temperature imaging on a Nikon A1 confocal microscope equipped with a ×40, 1.15 numerical aperture objective (water immersion; PlanApo Lambda DIC N2, Nikon). All optical datasets (79 nm lateral pixel resolution; 160 nm axial resolution) were deconvolved by 20 iterations of the Richardson–Lucy algorithm using NIS-Elements AR (version 5.30.02) for subsequent spatial alignment. The correlative target regions from each sample were processed for FIB-SEM via microwave-assisted processing and spatially aligned, as previously described[33]. The 3D electron microscopy was conducted using a Helios Nanolab G3 CX Dual-Beam FIB-SEM (FEI) microscope operating at 2 kV for back-scatter electron imaging (3.372 nm per pixel), with Gallium ion milling at 30 kV (10 nm per slice) using the Auto Slice and View (version 4;

FEI) software. The 3D FIB-SEM datasets were analysed via AIVE as previously reported[33]. using the following conditions. The Waikato Environment for Knowledge Analysis (WEKA v3.9) and 3D Trainable Weka Segmentation plugin for ImageJ were used to train a Random Forest classifier (250 iterations with 10% bagging, 149 image features, 12 features per branch, infinite depth and random seed of 1337) using 2,880,000 random training instances pooled equally from all six datasets. The classifier was trained to detect four classes of voxel: (1) extracellular material, (2) cytosolic material, (3) dense homogeneous non-membrane material and (4) membranes. An additional classifier was trained to detect cytoplasmic vesicles to aid their subclassification for AIVE (1,800,000 training instances, 149 image features and 1337 random seed); all other organelles were subclassified for AIVE manually, using the open-source Microscopy Image Browser. The signal-normalized inputs for AIVE were generated using an updated implementation of the Contrast Limited Adaptive Histogram Equalization approach, which accounted for voxel anisotropy in the XZ and YZ calculations. The membrane predictions, organelle subclasses and normalized voxel inputs were then processed via AIVE as described previously for the 3D segmentation of organellar membranes (iso-value of 64) in Paraview (version 5.9). Bulk mitochondrial morphology was calculated using the random forest model outputs for class 3 (homogeneous electron-dense material) to fill the mitochondrial interiors. Mitochondrial surface curvatures were calculated for bulk mitochondrial morphology using the Mean Curvature filter in Paraview (version 5.9). Rendering was conducted using Blender version 3.5 with the cycles engine.

### Immunocytochemistry

Control and TANGO2 (exons 3-9 *TANGO2* deletion) patient fibroblasts were seeded onto 13 mm diameter glass cover- slips and allowed to attach overnight. Cells were fixed with 4% paraformaldehyde at room temperature for 20 min, incubated in blocking buffer at room temperature for 45 min, followed by incubation with an anti-desmin antibody (67793-1-1 g, Proteintech, diluted 1:200) overnight at 4 °C. The following day cells were washed three times and incubated with secondary antibody (Alexa Fluor 488 Donkey anti-mouse IgG (H + L) Invitrogen A-21202, diluted 1:600) at room temperature for 1 h. Slides were imaged on a Nikon C2 confocal microscope equipped with a ×60, 1.4 numerical aperture objective (oil immersion; PlanApo VC DIC N2, Nikon).

### Statistical analyses

Values are means ± SD of biological replicates. A two-way Student's *t* test was used for most analyses assuming normal distribution unless otherwise stated, there was no blinding of data, at least three different biological replicates and three independent biological experiments were used for all presented data and the sample size is included in the figure legends. Randomization was not carried out and no animals were excluded from the analyses.

### Reporting summary

Further information on research design is available in the Nature Portfolio Reporting Summary linked to this article.

## Data availability

Genomic data are deposited to the Gene Expression Omnibus (GEO) as GSE276499. The mass spectrometry proteomics data have been deposited to the ProteomeXchange (http://proteomecentral. proteomexchange.org) via the PRIDE partner repository Project accession: PXD056037 and PXD055705 (for BioID experiments). Metabolomic data have been deposited to the Metabolomics workbench (ID:5903, https://doi.org/10.21228/M8BG16). *N*-glycome raw data files are available via GlycoPost[72] under identifier GPST000473. Source data are provided with this paper.

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

## Acknowledgements

This project was supported by grants from the NHMRC, ARC (CE200100029) and WACRF (to OR and AF). M.S, J.H, J.B and B.P were supported by UWA Postgraduate Scholarships. D.R was supported by the Zac Pearson Foundation. LCH is the Wesfarmers, UWA-VCCRI Chair in Cardiovascular Research; LCH acknowledges support from Woodside Energy and the NHMRC. We thank Dr Henrietta Cserne Szappanos for help with the echo acquisition, Dr Jasmin Browne for the TANGO2-GFP cells, the Monash Genome Modification Platform (MGMP), Monash Metabolic Phenotyping Platform and the Australian Phenomics Network (APN) node at Monash University for mouse embryo injections. We are grateful for the contribution of patient fibroblasts from Todd Andrew Hare. The Queensland Metabolomics and Proteomics (Q-MAP) node are supported by the Australian Government Department of Education through the National Collaborative Research Infrastructure Strategy (NCRIS), the Super Science Initiative, and the Collaborative Research Infrastructure Scheme and the Queensland Government. We thank the scientific and technical assistance of Microscopy Australia at the Centre for Microscopy, Characterisation & Analysis, The University of Western Australia.

## Author contributions

Conceptualization: O.R. and A.F. Investigation: M.S., L.A.H., B.P., D.L.R., S.V.F., S.J.S., T.M.c.C., K.L.P., J.A.E., J.H., F.R., T.E., J.E., L.C.H., N.H.P., E.S.X.M., B.S.P., O.R. and A.F. Visualization: M.S., L.A.H., D.L.R., S.J.S., B.S.P., O.R. and A.F. Funding acquisition: A.F., O.R., L.C.H., N.H.P. Project administration: A.F. Writing—original draft: A.F. Writing—review & editing: all authors.

## Competing interests

The authors declare no competing interests.
