## [Transparent Peer Review file · Nature Communications]

TANGO2 binds crystallin alpha B and its loss causes desminopathy

Corresponding Author: Professor Aleksandra Filipovska

Version 0:

Reviewer comments:

Reviewer #1

(Remarks to the Author)

Stentenbach and colleagues performed a detailed characterization of the role of TANGO 2 in cells and mice, which was thus far lacking. The amount of work and data generated in this study is staggering and it required some time to sift through it. TANGO2 deficiency results in adverse intermediate filaments, mitochondrial and metabolic remodeling, and ultrastructural abnormalities in striated and cardiac muscles. This is incredibly exciting work, and its prompt dissemination is warranted.

Specific comments:

1. It would be appropriate that the references regarding the role of desmin in the heart are updated with more recent reviews/articles.
2. Figure 1 a/h: why was the signal normalized to TOMM20 - wouldn't the Author expect for TOMM20 abundance to be different in TANGO -/- cells vs control?
3. Figure 1 b: the figure says that the signal was normalized to SDHA while the legend refers to VDAC1 – which one was it?
4. GAPDH is used consistently for loading controls. Being GAPDH a key metabolic enzyme, wouldn't the Authors expect for it to be changed? Do the proteomics data confirm that GAPDH is unaffected in Tango2 -/- vs controls? Given the abundance of the enzyme (esp. in the heart and muscle), its signal by Western blot can become saturated.
5. Figure 1 b-h: is it n=3 or =6?
6. What was the order chosen for the Methods? Perhaps it would be easier for the reader to follow if the Methods were listed according to the Results.
7. Figure 1i: was fragmentation measured manually or using an ImageJ macro? Perhaps the former would improve reproducibility (and eyesight..). Pictures have poor resolution/contrast in this version, could the Authors please provide higher res/larger images?
8. Could the Authors please highlight in yellow IF proteins in TAB S2? Perhaps the headings in the Table could be edited to reflect that the number refers to TANGO2 -/- vs Control? Perhaps create a sub-table with IF proteins only?
9. If space allows, could the Author expand on the significance of the changes in the fucose antenna for the heart and muscle?
10. In Figure 3b it looks like control sections are transversal and Tango2 -/- ones are longitudinal, perhaps the Authors could pick longitudinal sections for the controls as well?
11. Could the Authors please label Figure 5a? Perhaps the "cup-like shape could be highlighted in one sample/section (e.g. 19) as it is not always obvious.

12. The analysis of curvature in 5a is not easy to understand for the lay reader. Is it to be expected that the blue is confined to the center of the ROIs? Was there any difference in the perinuclear distribution of mitochondria between TANGO2 ^{-/-} and controls?

12. Could the Author please clarify how is a 3D rendering possible in 2D (Figure 5e)?

Minor

A. Shouldn't there be a space between "cells" and "(Extended)" (line 104)

Reviewer #2

(Remarks to the Author)

This tour de force aims to uncover the function of TANGO2. I have been asked to look specifically at the proteomics so will focus on that.

In general, the various applications of proteomics in this study seem well-done and the data support the conclusions. There were a few important details left out of the methods, which should be added. Most significantly, how many replicates were used? Were these biological replicates? Technical? This actually applies to most/all of the experiments described, beyond just the proteomics. "N" was missing in most places. Some other details would be helpful, too, such as whether imputation was used to fill in missing values or what kind of filtering might have been done on the protein lists (e.g., throw out those with >40% missing data, etc)

Reviewer #3

(Remarks to the Author)

While TANGO2 has been linked to a rare disease, its precise function in cells has not yet been elucidated. This manuscript uses a multi-omics approach to suggest a broad defect in cells that the authors explain is due to an interaction between TANGO2 and CRYAB, a heat shock protein that functions in desmin organization. The authors suggest that in the absence of TANGO2, desmin aggregates and this leads to changes in mitochondrial morphology, which they suggest explains the energy defect in patients with pathological TANGO2 mutations. They also use a mouse TANGO2 knockout model to support their hypothesis.

There is a lot of data in this manuscript. But I feel it is not developed enough to support their notion that TANGO2 regulates intermediate filament structure. Furthermore, the authors are either unaware or chose to ignore the recent data suggesting that B-vitamins, especially B5 and perhaps B9, have an effect on the TANGO2-associated phenotypes in patients. My specific comments and some more minor comments are below.

In my mind, the manuscript should have started with the yeast two-hybrid since the multi-omics didn't really suggest anything, but is used to support some later notions. Regarding yeast two-hybrid, no raw data is provided. The manuscript simply states that 121 out of 128 cDNA fragments identified were of CRYAB. What were the other 7? Having done yeast two-hybrid screens in the past, I find it rather odd that 95% of the clones were of a single protein. I am also not convinced that the CRYAB-TANGO2 interaction is real. It is rather weak (micromolar range) and the fact that they co-purify in a recombinant system is not that meaningful since many human proteins expressed in bacteria associate with chaperones. I did not see any type of control (another heat shock protein) or in vivo evidence of an interaction with the endogenous proteins). The BiolD performed was overexpressed CRYAB so the relevance is not clear. Is the R120G mutant of CRYAB properly folded? Why were pathological variants in TANGO2 not examined in their two-hybrid system? If none of these block the interaction with CRYAB, this calls the relevance into question.

The desmin aggregation assay (figure 4G) was quite hard to follow. As the temperature rises, I see desmin go into the pellet. Addition of CRYAB did not appear to have any effect on desmin at 37 or 44 degrees. TANGO2 addition with CRYAB had a very small effect. Nothing was quantified and it is unclear how many times this experiment was performed. I found this not a very compelling assay. A missing lane was desmin with only TANGO2. If that prevented desmin aggregation in the absence of CRYAB then the hypothesis is invalid.

I fail to see how vitamin B5 and vitamin B9 were not addressed in this work at all. How do the authors explain the prevention of metabolic crises reported due to these treatments (at least with a complete B complex administration or B5 alone). The discussion was an opportune time to bring this up but instead the discussion was nothing more than speculation on other ideas. Why did the authors not treat cells with B vitamins and examine the desmin network? What was the rationale for using HFD when TANGO2 metabolic crises are triggered by fasting or reduced carbohydrates? Mouse chow is rich in B vitamins so experiments on mice would require either supplementation with B vitamins or a specialized mix that is lacking or much lower in vitamins. Does B5, B9 or B-complex rescue any of the mouse phenotypes seen? B9 was recently reported to correct cardiac issues associated with TANGO2 loss in a cardiomyocyte model. That is an obvious experiment that was not performed.

In addition, throughout the manuscript no functional rescue is performed to convince the reader that effects they attribute to TANGO2 are due to TANGO2 dysfunction.

Along the lines of the vitamin and disease link, are the authors suggesting that TDD is a form of desminopathy? While some clinical features may be shared, in TDD the cardiac arrhythmia is triggered by diet or illness during a metabolic crisis. How does the CRYAB-TANGO2 interaction explain this, or the brain involvement which is a prominent feature in TDD patients (seizures, intellectual deficit). Similarities and differences between TDD and desminopathy were lacking and looking into them, I was not convinced that they were similar enough to suggest TDD is a form of desminopathy.

I also struggled to understand the glycomics. Many changes are related to the Golgi. No data was shown to suggest a change in Golgi morphology. The only view of the Golgi was shown in extended data figure 1 and neither the Golgi nor ER staining on those figures looked very good. As for the mitochondrial morphology, I am not an expert, but does simply changing its morphology affect oxphos? Is that known? Along these lines, the mitochondrial morphology changes were reported previously (PMID 32909282) but was not cited, so the changes shown in this study were not novel, though the methodology did suggest more details into those changes.

In general, figure legends are poorly written. It was difficult to understand the figures the way they were written. As just one example, the putative interaction between TANGO2 and CRYAB as shown in figure 4C is impossible to understand. Which is TANGO2 and which is CRYAB? The alpha-fold structure of TANGO2 suggests 2 beta sheets buried within the structure, yet the authors suggest that this is the interacting surface of TANGO2. That is (i) not clear in the figure and (ii) not very likely unless the TANGO2 protein opens up to expose the beta-sheets.

MINOR COMMENTS:

1. The TANGO2 community is using the acronym TDD (TANGO2 deficiency disorder) and not T2RD.
2. Why is a mitochondrial protein used as a loading control to show changes in other mitochondrial proteins?
3. I do not follow the logic that increased levels of glucose-6-phosphate and fructose-6-phosphate suggest increased glycolysis in the absence of oxphos. Doesn't the accumulation of glycolysis substrates suggest glycolysis is not being used?
4. Figure 1B: VDAC1 was used as a loading control but the figure suggests SDHA, and does not even show VDAC1.
5. Figure 1i: representative images of high, moderate, low and no mitochondrial fragmentation would be useful.
6. Why were CLA5 cells used to generate a TANGO2 knockout? What is the ploidy of such cells, and can that interfere with any of their analyses and conclusions?
7. The Excel files provided were not well described and it was difficult in some cases to know what I was looking at.
8. When the authors refer to the putative active site of TANGO2 being amino acids 25-27, what is its activity and what is this notion based on?
9. I did not find mention of which TANGO2 antibody was used in this study.
10. Metabolomics: N=5, but was this all analyzed at the same time? In my hands, repetition of metabolomics is tricky and I would need to see a repeat of these 5 samples to know that it is believable.

Reviewer #4

(Remarks to the Author)

The authors investigated the role of TANGO2 in both human cells and mice model, and demonstrated the changes in the mitochondrial and cytoplasmic proteomes, N-glycosylation and nucleocytoplasmic O-GlcNAcylation in the absence of TANGO2. Importantly, the mechanism that TANGO2 binds the small heat shock protein CRYAB has been identified, which further highlights the importance of TANGO2 in maintaining the cytoskeletal architecture that enables organelle networks, communication and function and have implications for future treatment options for patients with TANGO2 mutations. The work is of great importance and well-deigned. The data and results adequately support the conclusion. There is no additional evidence needed from my view. The methodology employed in this research has been well-described, which meet the expected standards.

There are few minor issues need to be addressed.

Line 65: please specify which cell(s)?

Line 591: Change "0-3 min—0 % B" to "0-3 min-0 % B"

Please correct the reference format, for example, line 930, change "Am J Hum Genet" to "Am. J. Hum. Genet."

Please supply the proposed structure of glycan in table S3.

Version 1:

Reviewer comments:

Reviewer #1

(Remarks to the Author)

No further actions required from this reviewer

Reviewer #2

(Remarks to the Author)

The authors have addressed my concern. Well done on an impressive study

Reviewer #3

(Remarks to the Author)

The revised manuscript addresses many of my initial concerns, particularly the rescue of the phenotypes by TANGO2 which I viewed as essential in any manuscript proposing to link a gene disruption to a phenotype. In that respect, the rescue still is significantly different from wild type in mitochondria function (figure S1c) which I think should be noted on line 87 by saying "significantly rescued" since it is still significantly reduced compared to wild type.

A few additional concerns remain, especially in light of a recent publication suggesting a different function for TANGO2 (PMID 40015245). This publication along with several others (PMIDs 32909282, 35197517, 18775783) suggest that TANGO2 is either at or inside mitochondria. The main conclusion of the present manuscript is that TANGO2, through a CRYAB interaction regulates intermediate filament structure, which affects mitochondria morphology, and that this in turn can explain the energy deficit in TDD patients. I think there needs to be a more explicit portion of the discussion section discussing these diverse roles for TANGO2. For example, how do the authors interpret the mitochondria association of TANGO2 in light of their work? Could it be a more localized/dramatic IF rearrangement near mitochondria? If TANGO2 is inside of mitochondria as recently suggested, how does this affect their interpretation of their data? Can the authors speculate on why, if TANGO2 interacts with CRYAB, is there rhabdomyolysis during a metabolic crisis? What compensates for this interaction outside of a metabolic crisis? All interesting discussion topics for this section.

Lastly, it would be important for the TANGO2 community to be aware that vitamin B5 does not have any effect on the phenotypes presented here, perhaps in the discussion section as "data not shown" if allowed by the journal, or as an extended figure.

Point-by-point response

Reviewer #1

Comments for the Authors

Stentenbach and colleagues performed a detailed characterization of the role of TANGO 2 in cells and mice, which was thus far lacking. The amount of work and data generated in this study is staggering and it required some time to sift through it.

TANGO2 deficiency results in adverse intermediate filaments, mitochondrial and metabolic remodeling, and ultrastructural abnormalities in striated and cardiac muscles. This is incredibly exciting work, and its prompt dissemination is warranted.

We thank the reviewer for their support of this work and their valuable advice and comments.

Specific comments:

1. It would be appropriate that the references regarding the role of desmin in the heart are updated with more recent reviews/articles.

We have updated the references as suggested, and if there are additional ones we have missed out please let us know which ones we should include.

2. Figure 1 a/h: why was the signal normalized to TOMM20 - wouldn't the Author expect for TOMM20 abundance to be different in TANGO -/- cells vs control?

We normalised to TOMM20 as this protein was not changing compared to all the others presented. We used TOMM20 for the (a) and (h) panels as it is a membrane protein like the membrane proteins analysed in these blots.

3. Figure 1 b: the figure says that the signal was normalized to SDHA while the legend refers to VDAC1 – which one was it?

We apologise for this error, the signal was normalised to SDHA in the blot as this protein was not changing and it is a protein localised to the matrix and matrix side of the inner membrane, the same as the location of the other DNA- and RNA-binding proteins analysed in this blot.

4. GAPDH is used consistently for loading controls. Being GAPDH a key metabolic enzyme, wouldn't the Authors expect for it to be changed? Do the proteomics data confirm that GAPDH is unaffected in Tango2 -/- vs controls? Given the abundance of the enzyme (esp. in the heart and muscle), its signal by Western blot can become saturated.

We used GAPDH as a loading control because it was not changing in any of the tissues by western blotting or proteomic analyses. TANGO2 affects intermediate filaments and cytoskeletal organisation therefore it is not surprising that it does not affect GAPDH levels.

5. Figure 1 b-h: is it n=3 or =6?

We apologise for this error it is n=6 for all immunoblots in this figure.

6. What was the order chosen for the Methods? Perhaps it would be easier for the reader to follow if the Methods were listed according to the Results.

We agree with this reviewer and have re-ordered the methods.

7. Figure 1i: was fragmentation measured manually or using an ImageJ macro? Perhaps the former would improve reproducibility (and eyesight..). Pictures have poor resolution/contrast in this version, could the Authors please provide higher res/larger images?

We measured the fragmentation manually and the individual figures of the manuscript have the higher resolution images, we apologize that compressing the manuscript and figures by the journal submission site lowers the resolution of individual images for review.

8. Could the Authors please highlight in yellow IF proteins in TAB S2? Perhaps the headings in the Table could be edited to reflect that the number refers to TANGO2 $-/-$ vs Control? Perhaps create a sub-table with IF proteins only?

We thank the reviewer for this suggestion and have made a sub-table for the IF proteins and edited the tab as suggested.

9. If space allows, could the Author expand on the significance of the changes in the fucose antenna for the heart and muscle?

We have included a discussion point on these changes in the revised manuscript as suggested.

10. In Figure 3b it looks like control sections are transversal and Tango2 $-/-$ ones are longitudinal, perhaps the Authors could pick longitudinal sections for the controls as well?

We have included longitudinal sections for the controls.

11. Could the Authors please label Figure 5a? Perhaps the “cup-like shape could be highlighted in one sample/section (e.g. 19) as it is not always obvious.

We have provided examples in Extended Data Figure 8, as suggested.

12. The analysis of curvature in 5a is not easy to understand from the lay reader. Is it to be expected that the blue is confined to the center of the ROIs? Was there any difference in the perinuclear distribution of mitochondria between Tango2 $-/-$ and controls?

Blue represents the convex shape of the mitochondrion and red is the concave. The FIB data did not include the nuclei, but the optical data clearly show the mitochondrial distribution in the corresponding cell. The optical data show that there were clear morphological changes in the mitochondria, but no alterations to the localisation of mitochondria with respect to the nucleus.

12. Could the Author please clarify how is a 3D rendering possible in 2D (Figure 5e)?

We apologize, the reviewer is correct the figure legend does not sufficiently describe the nature of the data. The data depict 3D sum projections of voxel values. Therefore the projections are based on 2D projections of 3D information. The revised figure legend now

states: *“Sum projected images of 3D voxel values showing surface filopodia detected in the control and TANGO2^{-/-} cells.”*

Minor

A. Shouldn't there be a space between “cells” and “(Extended” (line 104)
We have made this change.

Reviewer #2

This tour de force aims to uncover the function of TANGO2. I have been asked to look specifically at the proteomics so will focus on that.

We thank this reviewer for their support.

In general, the various applications of proteomics in this study seem well-done and the data support the conclusions. There were a few important details left out of the methods, which should be added. Most significantly, how many replicates were used? Were these biological replicates? Technical? This actually applies to most/all of the experiments described, beyond just the proteomics. "N" was missing in most places. Some other details would be helpful, too, such as whether imputation was used to fill in missing values or what kind of filtering might have been done on the protein lists (e.g., throw out those with >40% missing data, etc)

We used five biologically independent replicates for each experiment, and we have included this information in the figure legends. We have updated the methods to give some additional details requested by the reviewer, specifically that no imputation was used and proteins were considered significant if they had a false discovery rate less than or equal to 5% and an absolute log₂-fold change equal or greater than 0.6.

Reviewer #3

While TANGO2 has been linked to a rare disease, its precise function in cells has not yet been elucidated. This manuscript uses a multi-omics approach to suggest a broad defect in cells that the authors explain is due to an interaction between TANGO2 and CRYAB, a heat shock protein that functions in desmin organization. The authors suggest that in the absence of TANGO2, desmin aggregates and this leads to changes in mitochondrial morphology, which they suggest explains the energy defect in patients with pathological TANGO2 mutations. They also use a mouse TANGO2 knockout model to support their hypothesis.

There is a lot of data in this manuscript. But I feel it is not developed enough to support their notion that TANGO2 regulates intermediate filament structure. Furthermore, the authors are either unaware or chose to ignore the recent data suggesting that B-vitamins, especially B5 and perhaps B9, have an effect on the TANGO2-associated phenotypes in patients. My specific comments and some more minor comments are below.

We thank this reviewer for the time taken to assess the manuscript, we apologise for not being aware of the vitamin B studies for TANGO2 deficiency disorder, we would never ignore research and not cite it. We are grateful this has been brought to our attention and we have now included this aspect within the discussion of our work and carried out treatment experiments with vitamin B5 as discussed below.

In my mind, the manuscript should have started with the yeast two-hybrid since the multi-omics didn't really suggest anything, but is used to support some later notions. Regarding yeast two-hybrid, no raw data is provided. The manuscript simply states that 121 out of 128 cDNA fragments identified were of CRYAB. What were the other 7? Having done yeast two-hybrid screens in the past, I find it rather odd that 95% of the clones were of a single protein.

We reasoned that telling the story as it unfolded was an important message in uncovering the elusive role of TANGO2. We have included additional data on the yeast two-hybrid screens as requested. Having done yeast two hybrid work ourselves in the past, like this reviewer, we were amazed that so many hits came from the same protein. We note that these were all independent clones, as exemplified by the slightly different cDNA fragments sequenced in each clone, which greatly enhances the confidence in this result. Furthermore, we phenotypically validated this interaction using full-length CRYAB cloned into an independent expression plasmid to confirm the validity of the screen. In addition, to be certain of the interaction, we used four additional independent means to investigate the association between these proteins, and the consequences of TANGO2 loss on targets of these proteins, all of which are shown in Figure 4. In the revision, we have provided additional data for this reviewer on the other yeast two hybrid hits, which were artefacts and non-specific interactors, revealing that CRYAB was the key binding partner of TANGO2.

I am also not convinced that the CRYAB-TANGO2 interaction is real. It is rather weak (micromolar range) and the fact that they co-purify in a recombinant system is not that meaningful since many human proteins expressed in bacteria associate with chaperones. I did not see any type of control (another heat shock protein) or in vivo evidence of an interaction with the endogenous proteins).

We provide four different means to show that there is binding between these two proteins, an exhaustive number of different experiments to show this association, as well as effects in cells and tissues on the targets of CRYAB, desmin and vimentin. We show that desmin and vimentin are significantly reduced in both *TANGO2* knockout cells and in different tissues from the *Tango2* knockout mice. These data compellingly indicate that TANGO2 has a role in intermediate filament stability via its association with CRYAB. In addition, we find that TANGO2 does not associate or bind desmin, so that association with CRYAB is genuine. The microscale thermophoresis binding assay demonstrates binding in the micromolar range, however, this interaction between CRYAB and endogenous TANGO2 is likely to be meaningful given CRYAB's significant abundance in cells, so that this affinity would be biologically relevant, as well as the effects we see on CRYAB function in the absence of TANGO2 in multiple different assays.

The BioID performed was overexpressed CRYAB so the relevance is not clear. Is the R120G mutant of CRYAB properly folded? Why were pathological variants in TANGO2 not examined in their two-hybrid system? If none of these block the interaction with CRYAB, this calls the relevance into question.

All BioID experiments are only performed with genetically tagged proteins, and this is a commonly used technique that the reviewer may be using. We found that this mutant CRYAB protein is expressed in similar levels to the wild-type CRYAB shown by western blotting below. The TANGO2 mutation that we studied causes complete loss of TANGO2, and indeed this deletion of exons 3–9 is the most common patient mutation, therefore it is not possible to test this in the yeast two-hybrid system. Nevertheless, we carried out the test with

the pathological CRYAB mutation where we show reduced interaction with TANGO2, shown in Figure 4b, providing additional evidence of their association.

Response Figure 1 | Abundance of CRYAB mutant proteins in cells Western blotting of lysates expressing wild-type and knocked-in mutant CRYAB variants. GAPDH was used as a loading control (n=3 of each genotype).

The desmin aggregation assay (figure 4G) was quite hard to follow. As the temperature rises, I see desmin go into the pellet. Addition of CRYAB did not appear to have any effect on desmin at 37 or 44 degrees. TANGO2 addition with CRYAB had a very small effect. Nothing was quantified and it is unclear how many times this experiment was performed. I found this not a very compelling assay. A missing lane was desmin with only TANGO2. If that prevented desmin aggregation in the absence of CRYAB then the hypothesis is invalid.

The aggregation assay is a well-established method for measuring desmin aggregation in the literature (PMID: 26778558, PMID: 23530264, PMID: 28470624, PMID: 22096479, PMID: 15004226, PMID: 10559197). We provide quantitation of these experiments, which have at least four independent biological repeats, to further clarify the role of TANGO2 with the CRYAB holdase. TANGO2 does not associate with desmin and this is why this combination was not used in the aggregation assays.

I fail to see how vitamin B5 and vitamin B9 were not addressed in this work at all. How do the authors explain the prevention of metabolic crises reported due to these treatments (at least with a complete B complex administration or B5 alone). The discussion was an opportune time to bring this up but instead the discussion was nothing more than speculation on other ideas. Why did the authors not treat cells with B vitamins and examine the desmin network? What was the rationale for using HFD when TANGO2 metabolic crises are triggered by fasting or reduced carbohydrates? Mouse chow is rich in B vitamins so experiments on mice would require either supplementation with B vitamins or a specialized mix that is lacking or much lower in vitamins. Does B5, B9 or B-complex rescue any of the mouse phenotypes seen? B9 was recently reported to correct cardiac issues associated with TANGO2 loss in a cardiomyocyte model. That is an obvious experiment that was not performed.

We thank the reviewer for bringing this work to our attention and we have cited this work in the revised manuscript. We appreciate the experiments done with vitamin B, however, these are downstream of the molecular defect that we have identified here. In studies to date vitamin B5 has shown some promise, however, the mechanism of TANGO2 pathology has not been shown to be directly linked or caused by a vitamin B5 deficiency. We have identified here that TANGO2 assists CRYAB that affects the assembly of intermediate filaments and thereby impacts mitochondrial function by altering their dynamics via the cytoskeletal disorganization. Vitamin B5 is essential for the production of coenzyme A, which is critical for energy production and metabolism. Providing fuel for mitochondria that are in metabolic crisis may provide a benefit to patients, however, our study is focused on providing a mechanistic explanation for why patient mitochondria are in metabolic crisis – by

elucidating a previously unrecognized link between TANGO2, CRYAB and intermediate filaments.

Although our work provides a detailed understanding of the network up-stream of where vitamin B5 would have an effect, for this reviewer we have carried out vitamin B5 treatments following the exact methods according to the publication by Asadi *et al.* (2023). We provide the data here for this reviewer carried out by three different means. We cannot identify rescue in mitochondrial respiration, the association between desmin and CRYAB is not corrected by vitamin B5 treatment, nor are the Acetyl CoA levels, in fact the vitamin B5 treatments have worse effects on the *TANGO2*^{-/-} cells. Like this reviewer we also believe that identifying treatments of TDD is of utmost importance, however, this was not the goal of our present study. Given we do not observe rescue with vitamin B5 treatment and this is a new avenue that could be pursued using our models, we suggest these data are separate from our currently presented work and should be included in our future work. We are currently testing compounds that show promise in rescuing mitochondrial disorders by rescuing the mitochondrial network that we will apply to our TANGO2 models. Furthermore, will use our valuable models, established here, to dissect the effects of vitamin B5 on TDD that were discussed in Asadi *et al.* based on this reviewer's kind suggestions for follow up work that is outside of the scope of the current manuscript.

Response Figure 2 | Effects of vitamin B5 on respiration in *TANGO2*^{-/-} cells and CRYAB-desmin interactions (a) Mitochondrial respiration was measured in control and *TANGO2*^{-/-} cells treated with vehicle, 2 mM and 4 mM Vitamin B5 according to Asadi *et al.* (2023) using a high-resolution respirometer (n=8 per genotype). Three biological independent experiments were carried out. (b) Split-GFP protein-protein interaction assay for desmin and CRYAB binding in control and *TANGO2*^{-/-} cells treated with vehicle, 2 mM and 4 mM Vitamin B5 according to Asadi *et al.* (2023), n=3 per genotype and treatments. (c) Acetyl CoA levels in control and *TANGO2*^{-/-} cells treated with vehicle and Vitamin B5 according to

Asadi *et al.* (2023), n=5 per genotype and treatment. All values are means \pm SD * $p < 0.05$, ** $p < 0.01$ *** $p < 0.001$, **** $p < 0.0001$, Student's *t* test.

In addition, throughout the manuscript no functional rescue is performed to convince the reader that effects they attribute to TANGO2 are due to TANGO2 dysfunction.

Generally, rescue experiments are carried out to validate a newly identified or putative pathogenic mutation, whereas in our study the mutation was already known and the deletion of *TANGO2* in cells and *Tango2* in mice was deliberate and specific. In our work we sought to identify the role of TANGO2 and have provided three different models, knockout cells, knockout mice and patient fibroblasts, all showing the same defect, attributing its loss to the phenotypes in all three models. Nevertheless, we have provided rescue experiments (Extended Figure 1c and 1e) for this reviewer in the revised manuscript to show that loss of TANGO2 leading to mitochondrial OXPHOS and fragmentation defects are rescued by re-introducing TANGO2.

Along the lines of the vitamin and disease link, are the authors suggesting that TDD is a form of desminopathy? While some clinical features may be shared, in TDD the cardiac arrhythmia is triggered by diet or illness during a metabolic crisis. How does the CRYAB-TANGO2 interaction explain this, or the brain involvement which is a prominent feature in TDD patients (seizures, intellectual deficit). Similarities and differences between TDD and desminopathy were lacking and looking into them, I was not convinced that they were similar enough to suggest TDD is a form of desminopathy.

We propose that disruption of the cytoskeletal organization can affect the mitochondrial reticular network resulting in fragmentation, and reduced energy production and mitochondrial metabolism (shown by reduced OXPHOS and reduced metabolites of the CAC, including acetyl CoA). We would like to suggest that TDD can be a sub-class of desminopathy and that TANGO2 mutations may be screened in the clinic when a patient presents with a desminopathy that is not caused by desmin or CRYAB mutations. We are not trying to reclassify this disorder although we note that clinical presentations and disease classification can be tricky and not always black or white. For example, mutations in the PPA2 gene (although only very rare) resulting in cardiac arrest are also triggered by a metabolic crisis caused by viral infections, or by ingestion of alcohol and they are nevertheless classified as mitochondrial disorders, albeit they do not have all the features of the most common forms of mitochondrial disorders. These points are now considered in the Discussion of the revised manuscript.

I also struggled to understand the glycomics. Many changes are related to the Golgi. No data was shown to suggest a change in Golgi morphology. The only view of the Golgi was shown in extended data figure 1 and neither the Golgi nor ER staining on those figures looked very good. As for the mitochondrial morphology, I am not an expert, but does simply changing its morphology affect oxphos? Is that known? Along these lines, the mitochondrial morphology changes were reported previously (PMID 32909282) but was not cited, so the changes shown in this study were not novel, though the methodology did suggest more details into those changes.

Intermediate filament proteins, including desmin are glycosylated and this is not a defect that can impair Golgi morphology. Changes in glycosylation levels are a consequence of defects in intermediate filaments and loss of TANGO2 in our models. We show that TANGO2 is not

a Golgi protein therefore perhaps it is not surprising that Golgi morphology is not affected. Furthermore, we did not claim that mitochondrial fragmentation as a result of TANGO2 loss was new, but that it corroborates previous findings that TANGO2 loss affects OXPHOS function despite it not being a mitochondrial protein. The fact that TANGO2 affects intermediate filaments, which support the function, position and dynamics of mitochondria, explains why its loss causes morphological defects in mitochondria, fragmentation and OXPHOS defects.

In general, figure legends are poorly written. It was difficult to understand the figures the way they were written. As just one example, the putative interaction between TANGO2 and CRYAB as shown in figure 4C is impossible to understand,. Which is TANGO2 and which is CRYAB? The alpha-fold structure of TANGO2 suggests 2 beta sheets buried within the structure, yet the authors suggest that this is the interacting surface of TANGO2. That is (i) not clear in the figure and (ii) not very likely unless the TANGO2 protein opens up to expose the beta-sheets.

Apologies, but we are not clear why this reviewer has stated that in general the figure legends are poorly written as their opinion is not shared by the other three reviewers. We have further edited the figure legends with the specific comments from this reviewer to improve their clarity. The AlphaFold model of the TANGO2-CRYAB interaction illustrates a beta-sheet mediated interaction reminiscent of the interaction between CRYAB homo-oligomers, the beta sheet interaction occurs between the exposed edge strands of the beta sheets in each protein, therefore the TANGO2 protein doesn't need to "opens up to expose the beta-sheets". This structural model provides a potential insight into how TANGO2 may modulate CRYAB activity. Because this is only a model at this stage, we did not want to place too much emphasis on this finding. However, we are currently pursuing crystallography and cryo-EM experiments, which we hope will resolve this question for a future study.

MINOR COMMENTS:

1. The TANGO2 community is using the acronym TDD (TANGO2 deficiency disorder) and not T2RD.

We have made this change.

2. Why is a mitochondrial protein used as a loading control to show changes in other mitochondrial proteins?

Please see response 3 to reviewer 1's question.

3. I do not follow the logic that increased levels of glucose-6-phosphate and fructose-6-phosphate suggest increased glycolysis in the absence of oxphos. Doesn't the accumulation of glycolysis substrates suggest glycolysis is not being used?

Elevated levels of glucose-6-phosphate and fructose-6-phosphate indicate increased glycolysis, when oxidative phosphorylation is reduced or lost, as the accumulation of these metabolites suggests a higher flux through glycolysis in an effort to generate energy in the absence of aerobic metabolism. Increased levels of these metabolites as a result of reduced OXPHOS and increased glycolysis is a common feature of cancer cells and is not illogical.

4. Figure 1B: VDAC1 was used as a loading control but the figure suggests SDHA, and does not even show VDAC1.

We have corrected this error, we meant to say SDHA, please see response 3 to reviewer 1's question.

5. Figure 1i: representative images of high, moderate, low and no mitochondrial fragmentation would be useful.

This is an established method for evaluating mitochondrial morphology defects that is followed by experts in the field. Figure 2a in Lee *et al.* 2020 (PMID: 32576663) specifically shows examples of no fragmentation, low, moderate and high mitochondrial fragmentation.

6. Why were CLA5 cells used to generate a TANGO2 knockout? What is the ploidy of such cells, and can that interfere with any of their analyses and conclusions?

We have used CAL51 cells because they are diploid and thereby provide a more physiologically relevant model unlike the commonly used cell lines like HEK293T and HeLa, which are polyploid. We also have a mouse model and a patient cell model which corroborate the findings from the CAL51 cell work.

7. The Excel files provided were not well described and it was difficult in some cases to know what I was looking at.

The supplementary file lists the titles of the files and we have provided the titles in the files now.

8. When the authors refer to the putative active site of TANGO2 being amino acids 25-27, what is its activity and what is this notion based on?

We apologize, the reviewer is correct, this label is not clear. The mutation of amino acids 25-27 was devised to destabilize the main cleft in TANGO2, which may be important for its molecular interactions. We have corrected this text in the revised manuscript to clarify this point.

9. I did not find mention of which TANGO2 antibody was used in this study.

This information was provided in the original submission, and it can be found on line 685.

10. Metabolomics: N=5, but was this all analyzed at the same time? In my hands, repetition of metabolomics is tricky and I would need to see a repeat of these 5 samples to know that it is believable.

We have analyzed metabolomics data at different times with the same results each time, and the same is true for the proteomics and any other omics that we routinely use in our research projects. Our provider's protocols and extractions are consistent, and we have never come across different results because the samples have been run on different days. We routinely carry out these experiments with leaders in the field who specialize in metabolomics and would not provide data that could not be reproduced on different days.

Reviewer #4

The authors investigated the role of TANGO2 in both human cells and mice model, and demonstrated the changes in the mitochondrial and cytoplasmic proteomes, N-glycosylation and nucleocytoplasmic O-GlcNAcylation in the absence of TANGO2. Importantly, the mechanism that TANGO2 binds the small heat shock protein CRYAB has been identified, which further highlights the importance of TANGO2 in maintaining the cytoskeletal architecture that enables organelle networks, communication and function and have implications for future treatment options for patients with TANGO2 mutations. The work is of great importance and well-deigned. The data and results adequately support the conclusion. There is no additional evidence needed from my view. The methodology employed in this research has been well-described, which meet the expected standards. There are few minor issues need to be addressed.

Line 65: please specify which cell(s)?

Line 591: Change “0-3 min—0 % B” to “0-3 min-0 % B”

Please correct the reference format, for example, line 930, change “Am J Hum Genet” to “Am. J. Hum. Genet.”

Please supply the proposed structure of glycan in table S3.

We thank this reviewer for their support, and we have made all of the suggested edits as marked in the revised manuscript.

Point-by-point response

Reviewer #1

No further actions required from this reviewer

We thank the reviewer for their support of this work.

Reviewer #2

The authors have addressed my concern. Well done on an impressive study

We thank this reviewer for their support and kind words.

Reviewer #3

The revised manuscript addresses many of my initial concerns, particularly the rescue of the phenotypes by TANGO2 which I viewed as essential in any manuscript proposing to link a gene disruption to a phenotype. In that respect, the rescue still is significantly different from wild type in mitochondria function (figure S1c) which I think should be noted on line 87 by saying "significantly rescued" since it is still significantly reduced compared to wild type.

We have done this as suggested.

A few additional concerns remain, especially in light of a recent publication suggesting a different function for TANGO2 (PMID 40015245). This publication along with several others (PMIDs 32909282, 35197517, 18775783) suggest that TANGO2 is either at or inside mitochondria. The main conclusion of the present manuscript is that TANGO2, through a CRYAB interaction regulates intermediate filament structure, which affects mitochondria morphology, and that this in turn can explain the energy deficit in TDD patients. I think there needs to be a more explicit portion of the discussion section discussing these diverse roles for TANGO2. For example, how do the authors interpret the mitochondria association of TANGO2 in light of their work? Could it be a more localized/dramatic IF rearrangement near mitochondria? If TANGO2 is inside of mitochondria as recently suggested, how does this affect their interpretation of their data? Can the authors speculate on why, if TANGO2 interacts with CRYAB, is there rhabdomyolysis during a metabolic crisis? What compensates for this interaction outside of a metabolic crisis? All interesting discussion topics for this section.

We have addressed these aspects in the discussion further, which we believe are very important, particularly the localisation of TANGO2 as both our study and those of others, including the most recent study (PMID 40015245), have found TANGO2 not only localised to mitochondria but also outside of mitochondria. Our BioID work precisely aimed to examine how CRYAB is involved with mitochondrial proteins, as it has chaperone activity that can affect the stability of mitochondrial proteins. We have identified and confirmed previous interactors of CRYAB, such as VDAC1 (PMID: 27566162), and others that can affect mitochondrial function and through its association with TANGO2 can have an impact on mitochondrial proteins, like it has on intermediate filament proteins. Furthermore, the loss

of TANGO2 and its consequent effects on intermediate filaments affect mitochondrial morphology, which makes sense in light of the importance of intermediate filaments on mitochondrial function, distribution and shape (PMID: 27399781). This could also impact import or association with other organelles including lipid droplets, as suggested by PMID 40015245. This is evident from the pathology caused by TANGO2 loss in patients and in our mouse model that affects skeletal and heart function, including causing rhabdomyolysis (PMID 26805781). These points have been added to the revised Discussion (lines 287-291). Our current directions focus on understanding the nature of the interaction before and after metabolic crisis and the impact on mitochondrial metabolism.

Lastly, it would be important for the TANGO2 community to be aware that vitamin B5 does not have any effect on the phenotypes presented here, perhaps in the discussion section as "data not shown" if allowed by the journal, or as an extended figure.

We thank this reviewer for their nice suggestions and kind advice, unfortunately it is no longer a standard practise to refer to "data not shown", but the journal supports transparent review and these data will be available for the readers in those documents.